# Auto-nnU-Net: Towards Automated Medical Image Segmentation

**Jannis Becktepe**[†1,5]  **Leona Hennig**[1]  **Steffen Oeltze-Jafra**[2,4]  **Marius Lindauer**[1,3,4]

[1]Institute of AI, Leibniz University Hannover
[2]Peter L. Reichertz Institute for Medical Informatics, Hannover Medical School
[3]L3S Research Center
[4]CAIMed: Lower Saxony Center for AI & Causal Methods in Medicine
[5]Lamarr Institute for Machine Learning and Artificial Intelligence

**Abstract**  Medical Image Segmentation (MIS) includes diverse tasks, from bone to organ segmentation, each with its own challenges in finding the best segmentation model. The state-of-the-art AutoML-related MIS-framework nnU-Net automates many aspects of model configuration but remains constrained by fixed hyperparameters and heuristic design choices. As a full-AutoML framework for MIS, we propose Auto-nnU-Net, a novel nnU-Net variant enabling hyperparameter optimization (HPO), neural architecture search (NAS), and hierarchical NAS (HNAS). Additionally, we propose Regularized PriorBand to balance model accuracy with the computational resources required for training, addressing the resource constraints often faced in real-world medical settings that limit the feasibility of extensive training procedures. We evaluate our approach across diverse MIS datasets from the well-established Medical Segmentation Decathlon, analyzing the impact of AutoML techniques on segmentation performance, computational efficiency, and model design choices. The results demonstrate that our AutoML approach substantially improves the segmentation performance of nnU-Net on 6 out of 10 datasets and is on par on the other datasets while maintaining practical resource requirements. Our code is available at `https://github.com/automl/AutoNNUnet`.

## 1 Introduction

Machine learning (ML) plays a key role in modern healthcare, enabling accurate diagnoses (Fauw et al., 2018; Bernard et al., 2018; Khan et al., 2023; Wang et al., 2024), early cancer detection (Cao et al., 2023), and scientific discovery (Falk et al., 2018). Medical image segmentation (MIS) aims to identify anatomical structures in medical scans but is challenging due to datasets variations, class imbalances, and task-specific constraints (Litjens et al., 2017; Isensee et al., 2020a; Ali et al., 2024).

Self-configuring methods reduce the need for manual tuning by adapting models for a given dataset (Ali et al., 2024). nnU-Net (Isensee et al., 2020a) has emerged as a state-of-the-art framework that automatically configures U-Net-based architectures to achieve strong segmentation performance. However, nnU-Net surprisingly relies on some fixed hyperparameters and manually designed heuristics, which limit flexibility and may not always yield optimal results across datasets (Bergstra et al., 2012; Quinton et al., 2024).

In this work, we leverage automated machine learning (AutoML) (Hutter et al., 2019) to address these challenges and perform a large-scale study on the impact of AutoML on MIS. We introduce Auto-nnU-Net, a novel variant of nnU-Net that integrates AutoML to enable hyperparameter optimization (HPO) and neural architecture search (NAS) for nnU-Net. By combining PriorBand (Mallik et al., 2023) with multi-objective optimization (Karl et al., 2023), we introduce Regularized PriorBand for Joint Architecture and Hyperparameter Search (JAHS) (Awad et al., 2023), addressing the growing interest in resource efficiency in MIS as highlighted by recent work (Rayed et al., 2024). Our

---

[†]Work was conducted at Institute of AI, Leibniz University Hannover.

study evaluates Auto-nnU-Net on the Medical Segmentation Decathlon (MSD) datasets (Simpson et al., 2019; Antonelli et al., 2022), providing insights into the impact of optimization strategies, hyperparameter importance, and dataset characteristics. Notably, unlike most studies on AutoML for MIS (Ali et al., 2024), we report results for all ten MSD datasets, providing a more thorough assessment of generalizability and robustness across diverse medical segmentation challenges.

In this work, we make the following contributions:

1. **Auto-nnU-Net for AutoML-driven MIS**. We propose a novel framework that automates key design decisions in nnU-Net for flexible and structured HPO and NAS.

2. **Efficient optimization with Regularized PriorBand**. We introduce Regularized PriorBand, which incorporates training runtime as an optimization objective to reflect real-world constraints, where limited resources and frequent retraining make efficient training essential. It selects slower models only if they improve accuracy, and inherently yields trade-off solutions.

3. **Extensive evaluation across all ten MSD datasets**. We analyze the impact of AutoML on segmentation accuracy, including hyperparameter importance and dataset transferability, enabling a deeper understanding of generalization behavior and guiding the design of more robust, efficient models across diverse medical imaging tasks.

## 2 Background on Image Segmentation

Following Szeliski (2022), *semantic segmentation* refers to partitioning an image into regions associated with specific classes. We use the term *image segmentation* interchangeably to describe this task, where each pixel is labeled to enable structured analysis of visual data.

*Medical image segmentation* (MIS) involves partitioning medical images, e.g., magnetic resonance imaging (MRI) or computer tomography (CT) scans, to identify areas of interest, incl. organs or potentially malicious structures such as tumors (Antonelli et al., 2022). In practice, automated segmentation assists clinicians by accurately identifying critical areas for patient treatment (Liang et al., 2019). Recent MIS datasets focus on foreground classes, treating the background as a single, excluded class (Menze et al., 2015; Heller et al., 2019; Simpson et al., 2019; Antonelli et al., 2022). Unlike natural image segmentation, MIS faces challenges like limited availability of training data, class imbalances, small or branching anatomical structures, weak boundaries, and variable intensity distributions. The segmentation of 3D images consisting of *voxels* (volume pixels) from MRI and CT scans further increases the segmentation complexity and computational demands of MIS (Isensee et al., 2020a; Ali et al., 2024).

## 3 Related Work

In this section, we review previous work on core components of Auto-nnU-Net: self-configuring segmentation frameworks, HPO, NAS, and multi-objective optimization for MIS. Our work aims to unify them into a comprehensive AutoML framework tailored to the task at hand.

**nnU-Net**. Self-configuring frameworks address the challenge of designing and tuning MIS models for a given task and dataset. Building on U-Net's success (Ronneberger et al., 2015), Isensee et al. (2020a) introduced nnU-Net, which optimizes an U-Net for a given task. Like CASH (Thornton et al., 2013), it jointly selects training hyperparameters and the final model or ensemble for inference. nnU-Net achieves this by leveraging dataset *meta-features* common in AutoML (Vanschoren, 2019).

We focus on the self-configuration mechanism of nnU-Net, omitting pre- and post-processing steps. The pipeline consists of three phases: (i) **Experiment planning**, where rule-based hyperparameter selection leverages dataset properties; (ii) **Training**, where 2D U-Net, 3D U-Net, and, if needed, a 3D U-Net cascade are trained using 5-fold cross-validation; (iii) **Inference**, where the best-performing model or ensemble is selected based on validation scores. nnU-Net relies on three

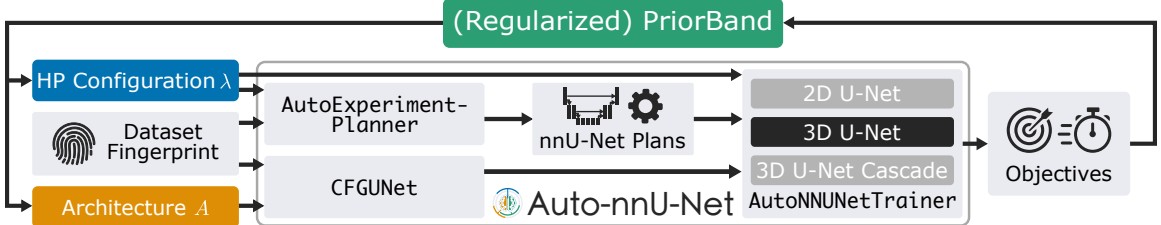

Figure 1: Overview of the Auto-nnU-Net framework: Given a hyperparameter configuration $\lambda$, architecture $A$, and dataset fingerprint, the `AutoExperimentPlanner` and `CFGUNet` generate nnU-Net training plans and model architecture, respectively. The `AutoNNUNetTrainer` then trains the selected model, providing runtime and validation score as objectives to the PriorBand optimizer. For details, see Appendix C.1. For more details, see Appendix C.1.

types of hyperparameters: (i) **Fixed** (e.g., learning rate, optimizer, loss function); (ii) **Rule-baesd** (e.g., preprocessing, network topology); (iii) **Heuristic** (e.g., ensemble selection, post-processing).

**Hyperparameter Optimization (HPO) for MIS**. In general, there is little work on HPO for MIS. Yang et al. (2019) propose reinforcement learning for optimizing the data augmentation and learning rate of a 3D segmentation model. Quinton et al. (2024) apply HPO to different models, including nnU-Net, by subsequently performing Bayesian optimization (BO) for three groups of hyperparameters: (i) patch size, (ii) data pre-processing and augmentation, and (iii) loss function and optimizer.

**Neural Architecture Search (NAS) for MIS**. Various NAS methods have been adapted for MIS. Several approaches build on DARTS (H. Liu et al., 2019) and apply differentiable NAS to encoder-decoder-based MIS models (Weng et al., 2019; Zhu et al., 2019; Y. He et al., 2021). Another approach adopts a coarse-to-fine strategy for U-Net-shaped networks, first optimizing the overall topology before refining cell-level operations (Q. Yu et al., 2020). Evolutionary and graph-based NAS methods have also been proposed for MIS, using genetic algorithms (Hassanzadeh et al., 2020; Khouy et al., 2023; C. Yu et al., 2023) and graph representations of architectures that are optimized or expanded during training to reduce search time and improve flexibility (R. Liu et al., 2023; Qin et al., 2023).

**Multi-Objective Optimization and Joint HPO and NAS for MIS**. Prior work has applied multi-objective NAS to MIS to balance performance and resource constraints (Baldeon-Calisto et al., 2020; Lu et al., 2022), but without tuning hyperparameters. Yang et al. (2021) combine HPO and NAS via surrogate models to optimize U-Net configurations, but do not consider resource efficiency.

## 4 Auto-nnU-Net for MIS

In this work, we present Auto-nnU-Net, a novel approach that integrates AutoML methods into nnU-Net. Furthermore, we introduce Regularized PriorBand to enable efficient Joint Architecture and Hyperparameter Search (JAHS) (Awad et al., 2023) in Auto-nnU-Net.

### 4.1 Integrating AutoML Methods into nnU-Net

nnU-Net provides robust segmentation pipelines, including data pre-processing, experiment planning, training, and inference. However, its fixed and rule-based hyperparameters limit configurability. To address these limitations, we propose **Auto-nnU-Net**, which enhances nnU-Net with flexible experiment planning and training. Figure 1 shows an overview of our framework. Unlike nnU-Net, Auto-nnU-Net takes hyperparameter and architecture configurations as inputs, enabling JAHS. It returns both generalization error and training runtime to allow the optimization process to account for both segmentation performance and computational efficiency.

## 4.2 Regularized PriorBand for Efficient Joint HPO and NAS

Building upon the flexible Auto-nnU-Net framework, we further enhance the optimization process using **Regularized PriorBand**. In this section, we describe how we extend PriorBand (Mallik et al., 2023) from HPO to JAHS. Given that nnU-Net requires considerable training cost and provides a strong prior configuration, we aim to incorporate this knowledge into the optimization process to improve its efficiency. To achieve this, we leverage PriorBand (Mallik et al., 2023), a multi-fidelity HPO method specifically designed to integrate prior knowledge into the optimization of computationally expensive deep learning models. It enhances exploration by combining random, prior-based, and incumbent-based sampling strategies to dynamically adjust as the optimization progresses. Random sampling explores the search space, prior-based sampling leverages expert knowledge, and incumbent-based sampling refines the current best-performing configuration.

We extend the HPO search space of PriorBand by encoding architectures within a unified configuration space (Zela et al., 2018). However, exploring larger models introduces computational challenges. While increased model size can enhance accuracy, it also raises optimization costs and prolongs training. We consider training runtime as an optimization objective to better reflect the practical constraints of medical environments, where computational resources are often limited and large-scale training may be infeasible (Rayed et al., 2024). Dataset heterogeneity — due to technical factors (e.g., scanners, protocols) and anatomical variability (e.g., organ shape, number of structures) — often necessitates repeated fine-tuning or model adaptation. In such continual learning scenarios, where retraining is recurring and costly (Isensee et al., 2020a; Wagner et al., 2024), efficient training is essential. Privacy constraints often prevent centralized access to patient data, requiring localized or federated retraining when new data becomes available, which further emphasizes the importance of minimizing training costs (Wagner et al., 2024).

The central idea of Regularized PriorBand is that larger models should only be considered if they contribute to accuracy improvement. Ultimately, the goal remains to optimize for accuracy, ensuring that the best-performing configurations are not discarded in favor of resource-constrained choices. An overview of Regularized PriorBand is provided in Algorithm 2 in Appendix C.

**Selection Strategy in Successive Halving**. A key adaptation in Regularized PriorBand involves modifying the configuration selection strategy in the Successive Halving (SH) subroutine (Jamieson et al., 2016). In the standard SH approach, configurations for the next higher budget are selected based on their cost, with the configurations exhibiting the lowest cost being prioritized for evaluation. However, when optimizing for both accuracy and training runtime, we must consider a vector of costs rather than a single scalar to account for both objectives.

To integrate both accuracy and runtime, we modify the selection process by employing non-dominated sorting and crowding distance sorting as proposed by Deb et al. (2002) and similar to recent work (Izquierdo et al., 2021; Schmucker et al., 2021; Awad et al., 2023). After evaluating configurations at the current budget, we apply non-dominated sorting to group configurations into fronts. To favor diverse solutions, we rank the configurations by their crowding distance within each front. From these sorted fronts, we select the top $k$ configurations for evaluation at the next higher budget, beginning with the first front and continuing until $k$ configurations are chosen. If two configurations have equal crowding distances, the selection prioritizes accuracy. This guarantees that the configuration with the highest accuracy is always promoted.

**Incumbent Selection**. In Regularized PriorBand, the final incumbent configuration is selected based on accuracy, without considering runtime. However, to enable incumbent-based sampling throughout the optimization, we incorporate both accuracy and runtime. The selection is limited to configurations on the approximated Pareto front, ensuring a balance between the two objectives. To choose the incumbent for the local search, we compute the area spanned by the normalized objective costs and select the configuration that maximizes this area, facilitating the exploration of trade-offs between accuracy and runtime.

## 5 Experimental Setup

Based on our Auto-nnU-Net framework, we conduct the most comprehensive study on AutoML for MIS to date, being equivalent to approximately 60 k GPU hours and 10 964 kg $CO_2$ equivalents (see Appendix A for more details). Instructions to reproduce all experiments, results, and visualizations can be found in our GitHub repository at `https://github.com/automl/AutoNNUnet`. All experiments are performed using 5-fold cross-validation. See Appendix D.3 for more details.

### 5.1 Datasets

To ensure a comprehensive evaluation of our methods, we use the Medical Segmentation Decathlon (MSD) (Simpson et al., 2019; Antonelli et al., 2022), a benchmark of ten MIS datasets designed to capture diversity across clinical tasks, imaging modalities, and data characteristics (see Appendix D.1). The MSD uses the Dice Similarity Coefficient (DSC) (Dice, 1945), an effective metric for evaluating MIS methods (Zijdenbos et al., 1994). The DSC measures the overlap between ground truth (X) and prediction (Y) as $DSC(X, Y) = \frac{2|X \cap Y|}{|X|+|Y|}$ ranging from 0 (no overlap) to 1 (perfect overlap).

### 5.2 Baselines

In our experiments, we aim to investigate how AutoML methods can improve the segmentation performance of current MIS methods. Our first baseline is the 3D U-Net of the nnU-Net framework, particularly its default configurations (i) *Conv*, (ii) *ResM*, and (iii) *ResL*. Additionally, we evaluate MedSAM2 (Ma et al., 2024a), a foundation model-based approach for MIS. Unlike nnU-Net, MedSAM2 leverages large-scale pre-training and serves as a state-of-the-art competitor to our approach. We leverage the pipeline proposed by the authors to finetune MedSAM2 on each individual MSD dataset for 100 epochs, which is roughly equivalent to the training runtime of the most expensive nnU-Net configuration on D01, the dataset with the highest training runtime.

### 5.3 Evaluation of Auto-nnU-Net

This section outlines the experimental setup for evaluating our Auto-nnU-Net approach. Auto-nnU-Net uses Regularized PriorBand to incorporate the *Training Runtime* objective alongside *1 - DSC* into the optimization to prefer more efficient models at equal performance. For the PriorBand optimizer, we rely on the setup proposed by Mallik et al. (2023) (see Appendix D.2.2), on one random seed due to the extensive computational resources required otherwise.

The Auto-nnU-Net search space includes both regular hyperparameters, which define training and configuration settings (e.g., learning rate and data augmentation), and architectural hyperparameters, which govern the network structure (e.g., encoder type and dropout rate). This JAHS-search-space formulation (Bansal et al., 2022) enables simultaneous tuning of training dynamics and model capacity. The full Auto-nnU-Net search space is given in Table 4 and as a combination of the HPO and NAS spaces. Details on the hyperparameters are stated in Appendix D.2.3.

### 5.4 Ablation Variants

To assess the contribution of different components within the Auto-nnU-Net framework, we define two ablation variants that isolate or modify parts of the JAHS search space:

**HPO using PriorBand.** In this variant, we disable the architectural search of Auto-nnU-Net and optimize only the regular hyperparameters, using PriorBand (Mallik et al., 2023)) without the added regularization from Auto-nnU-Net, to minimize *1 - DSC*. By excluding architectural hyperparameters, this ablation isolates the effect of tuning configuration choices and helps to quantify the performance gains attributable solely to hyperparameter optimization when the network architecture is fixed to the nnU-Net default. In Table 4 (top), the ranges and sets of possible values for each hyperparameter are defined.

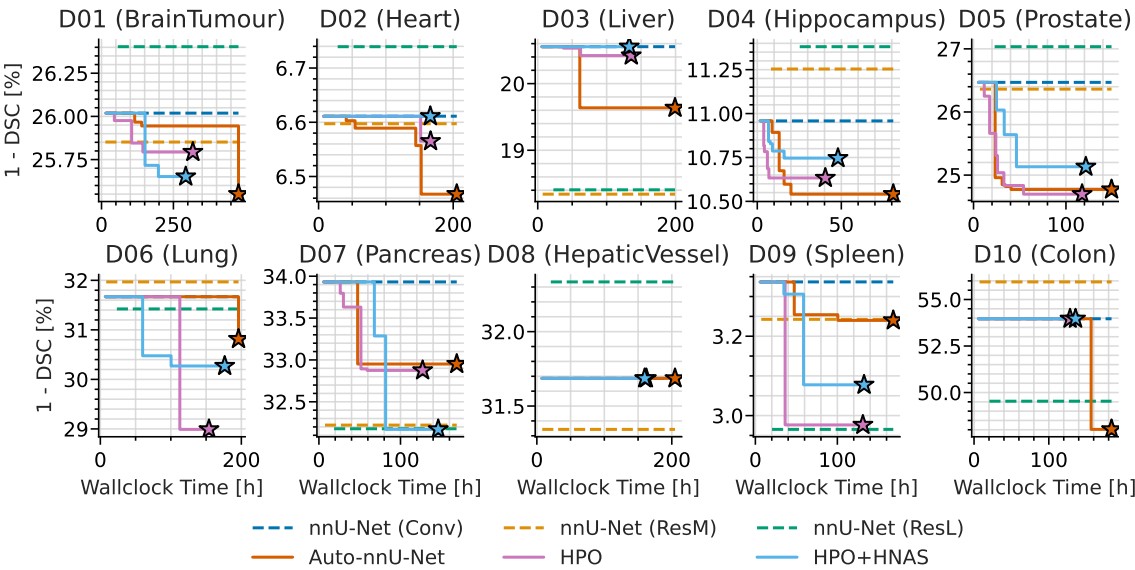

Figure 2: Incumbent performance of nnU-Net, Auto-nnU-Net, and Auto-nnU-Net ablations over time. Detailed results for each dataset are stated in Appendix E. Final validation DSCs are stated in Table 6 in Appendix E. We exclude MedSAM2 as it fails to achieve the performance of nnU-Net on nine out of ten datasets (see Table 6).

**Hierarchical NAS for U-Nets**. While Regularized PriorBand enables JAHS as in Auto-nnU-Net, its search space is limited to predefined modifications. To explore a broader range of U-Net architectures while maintaining efficiency and feasibility, we introduce a hierarchical NAS (HNAS) search space, leveraging context-free grammars (CFG) (Schrodi et al., 2023) to systematically refine and optimize U-Net structures (see Appendix C.3.1). Our approach preserves nnU-Net's default configurations while introducing flexible topological and cell-level design choices. We apply prior-based sampling with Regularized PriorBand, modeling CFG production rules as categorical and integer hyperparameters to integrate smoothly with existing nnU-Net components (see Appendix C.3.2).

## 6 Results

In this section, we present empirical results demonstrating the effectiveness of Auto-nnU-Net across the MSD datasets. We evaluate segmentation accuracy, efficiency, and configuration transferability, comparing against baselines and ablations. To gain a deeper understanding of the underlying optimization behavior, we additionally analyze the importance of individual hyperparameters.

### 6.1 Auto-nnU-Net Results

First, we discuss the optimization progress of Auto-nnU-Net across MSD datasets. Figure 2 shows the Auto-nnU-Nets incumbent *1 - DSC* over time compared to the default nnU-Net baselines. Except for D08, where the DSC matches, Auto-nnU-Net outperforms nnU-Nets convolutional default. Notably, for D04, Auto-nnU-Net identifies the incumbent configuration faster than training nnU-Net (ResL), highlighting its efficiency over computationally expensive models.

The final validation results, including MedSAM2, are stated in Table 6 in Appendix E. We exclude MedSAM2 from the cost-over-time comparison as it underperforms nnU-Net on nine out of ten datasets, only outperforming it on D10. We hypothesize that MedSAM2s performance on D10 is due to its requirement of a bounding box prompt from the ground truth mask, which helps with detecting small target regions. While this approach benefits detection tasks, it requires additional annotation, whereas our method works without such supervision.

| | nnU-Net | | | Auto-nnU-Net | Ablation | |
| --- | --- | --- | --- | --- | --- | --- |
| | Conv | ResM | ResL | | HPO | HPO+HNAS |
| D01 | 61.34 ± 24.3 | 61.21 ± 24.0 | 61.12 ± 24.4 | **61.58 ± 24.3** | 61.06 ± 24.2 | 57.31 ± 25.3 |
| D02 | 93.33 ± 1.5 | 93.36 ± 1.5 | 93.03 ± 1.8 | **93.46 ± 1.4** | 93.28 ± 1.5 | 93.35 ± 1.5 |
| D03 | 85.36 ± 13.0 | 86.33 ± 11.6 | **86.66 ± 11.4** | 85.91 ± 12.0 | 85.36 ± 13.1 | 85.68 ± 12.4 |
| D04 | 89.43 ± 3.8 | 89.22 ± 3.8 | 89.10 ± 3.7 | **89.75 ± 4.0** | 88.34 ± 3.9 | 88.99 ± 3.8 |
| D05 | 80.91 ± 7.0 | 79.98 ± 7.0 | 80.65 ± 6.5 | **82.29 ± 5.9** | 81.95 ± 6.4 | 77.96 ± 9.3 |
| D06 | 67.14 ± 30.6 | 62.44 ± 34.6 | **70.52 ± 26.2** | 68.78 ± 26.9 | 69.70 ± 26.7 | 69.83 ± 21.9 |
| D07 | 64.70 ± 20.5 | 66.45 ± 21.0 | **66.68 ± 21.3** | 65.23 ± 20.8 | 66.38 ± 19.7 | 65.63 ± 19.7 |
| D08 | 68.37 ± 19.0 | **68.48 ± 18.8** | 68.35 ± 19.2 | 68.23 ± 19.2 | 68.12 ± 19.3 | 68.43 ± 19.1 |
| D09 | 97.23 ± 1.2 | 97.11 ± 1.3 | 94.93 ± 10.7 | 97.11 ± 1.4 | **97.34 ± 1.0** | 96.62 ± 1.3 |
| D10 | 52.96 ± 35.4 | 48.26 ± 38.2 | 50.36 ± 36.2 | **58.05 ± 32.0** | 47.20 ± 37.8 | 50.88 ± 36.7 |
| **Mean** | 76.08 ± 15.6 | 75.28 ± 16.2 | 76.14 ± 16.1 | **77.04 ± 14.8** | 75.87 ± 15.4 | 75.47 ± 15.1 |

Table 1: Mean ± standard deviation of the DSC [%] for the MSD test set obtained through the official submission platform for all datasets (rows) and approaches (columns). Metrics are computed over all test set instance DCSs per dataset. The best-performing method per dataset is highlighted in **bold**. Notably, as MedSAM2 requires access to the ground truth segmentations to generate prompts, the model cannot be evaluated on unlabeled data.

To assess performance on unseen data, we evaluate the MSD test set results. Table 1 presents the final test set DSC [%] for all approaches, excluding MedSAM2, which requires ground truth segmentations that are unavailable for the MSD. Consistent with the validation results, Auto-nnU-Net achieves the highest average DSC (77.04%). Our method surpasses all nnU-Net baselines and demonstrates strong generalization, ranking best on five out of ten datasets.

Figure 3 shows qualitative results for the best validation case in D01. All methods correctly segment the *Edema* class but struggle with the other foreground classes. Notably, MedSAM2 fails to segment the Enhancing tumor voxels within the *Non-enhancing tumor* region and over-segments the *Non-enhancing tumor* class. In contrast, Auto-nnU-Net captures fewer voxels of both the *Non-enhancing tumor* and *Enhancing tumor* classes. These results highlight that MedSAM2 over-segments the target regions, while other methods under-segment them.

Regularized PriorBand inherently balances accuracy and training runtime during optimization (Section 4.2). Figure 4 compares the Pareto fronts of Auto-nnU-Net, its ablations (HPO, HPO+HNAS), and baselines (nnU-Net, MedSAM2) on D03 and D04, illustrating objective trade-offs. On D03, Auto-nnU-Net and HPO+HNAS reveal clear accuracy-runtime trade-offs, while nnU-Net (ResM) achieves high DSC with low runtime. On D04, Auto-nnU-Net and HPO+HNAS outperform all nnU-Net variants in accuracy and significantly reduce training time — HPO+HNAS cuts runtime by a factor of 26. MedSAM2 underperforms on both datasets. These results demonstrate Auto-nnU-Net's ability to jointly optimize accuracy and efficiency.

**Ablation Results.** We discuss the HPO and HPO+HNAS ablations of Auto-nnU-Net in Figure 2 and Table 1. HPO outperforms nnU-Net (Conv) on all datasets except D08 and D10, while Auto-nnU-Net shows similar improvements, excelling on all but D08. HPO+HNAS surpasses nnU-Net (Conv) on six datasets and generally optimizes more efficiently than both Auto-nnU-Net and HPO for some datasets, like D01. However, compared to Auto-nnU-Net, both HPO and HPO+HNAS exhibit lower DSCs on the test set, suggesting reduced robustness to unseen data. For D01, despite similar validation DSCs, HPO+HNAS underperforms relative to Auto-nnU-Net, indicating greater sensitivity to unseen data and effectiveness of encoding neural architectures as hyperparameters.

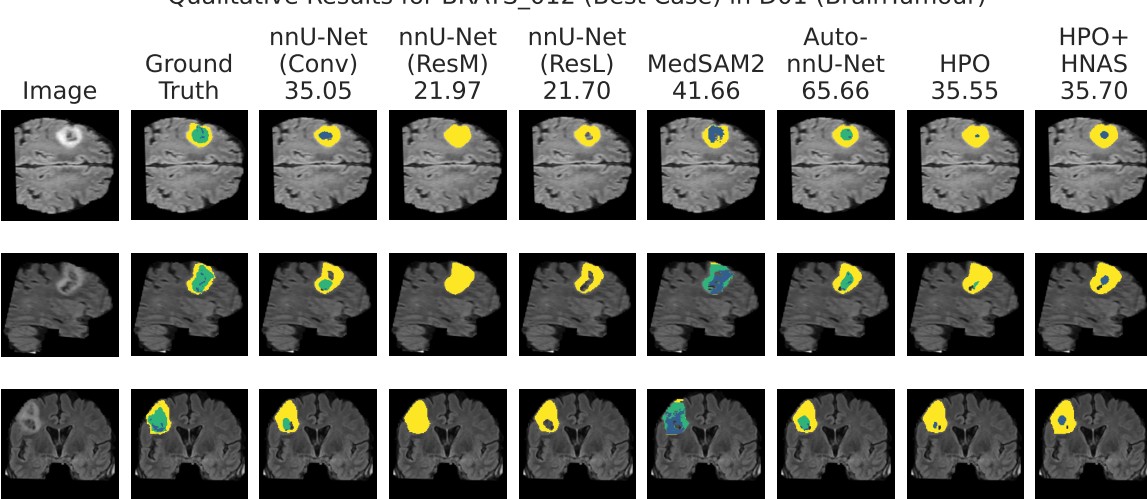

Figure 3: Qualitative segmentation results for D01. Columns show the input image, ground truth mask, and method predictions, with colors denoting foreground classes. Numbers below method names indicate DSC scores [%] for this example. Each row shows a slice of the 3D volume along one axis. As the 4D volume is an mp-MRI scan, the first parameter setting is used to extract a 3D volume. Additional results are in Appendix E.

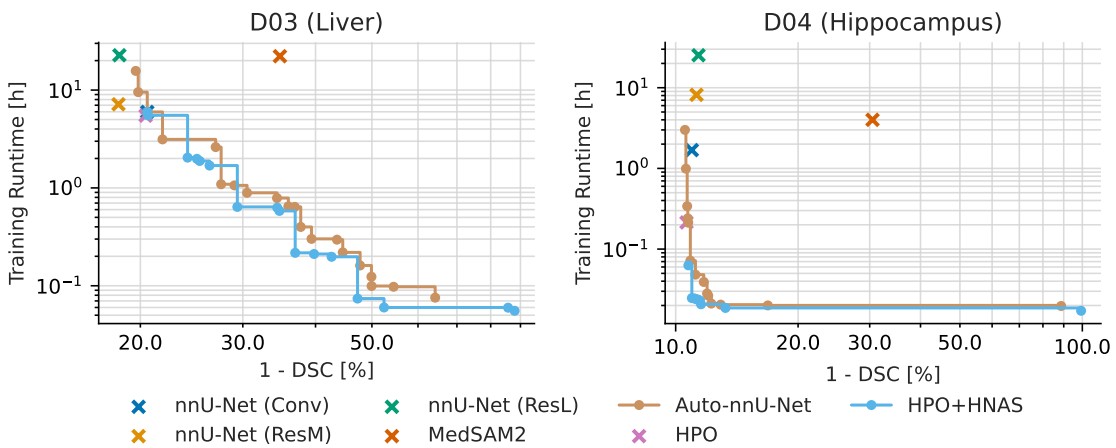

Figure 4: Pareto fronts of Auto-nnU-Net and HPO+HNAS for D03 **(left)** and D04 **(right)** compared to the baselines and HPO results. Additional results are stated in Appendix E.

### 6.2 Analysis of Hyperparameter Importance in Auto-nnU-Net

To assess the impact of individual hyperparameters on accuracy, we use functional ANOVA (fANOVA) (Hutter et al., 2014) to estimate their global importance across the configuration space by decomposing performance variance into contributions from each hyperparameter and their interactions. Figure 14 shows their importance across datasets. Key hyperparameters such as *Foreground Oversampling*, *Initial Learning Rate*, and *Momentum (SGD)* are consistently influential, though importance varies notably between datasets. This highlights the value of AutoML over fixed settings, as used in the original nnU-Net. In contrast, hyperparameters like *Encoder Type* and *Normalization* show low importance and may not require further optimization.

### 6.3 Transferring Incumbent Configurations across Datasets

A key question in evaluating dataset influence on AutoML is whether an optimized configuration for one dataset generalizes to others. We analyze the transferability of Auto-nnU-Nets incumbents, excluding D08 where it does not outperform the default, resulting in a $9 \times 10$ matrix (Figure 13). In half of the datasets, the tailored incumbent does not achieve the highest DSC, with the D02 configuration showing the largest gain (+2.78%) on D05, while the D03 incumbent achieves the highest DSC on four datasets. However, its lower performance on D05 prevents it from surpassing the nnU-Net default on average. These results suggest that configurations can transfer across datasets — e.g., D03 performs best on D03, D08–D10 — but others, like D04 and D05, generalize poorly, particularly to D06–D10. This highlights the potential of meta-learned HPO for improved transferability in MIS (Feurer et al., 2015; Wistuba et al., 2015; Schilling et al., 2016).

## 7 Conclusion and Future Work

In this work, we proposed Auto-nnU-Net, an automated framework for medical image segmentation that combines nnU-Net with structured HPO and NAS. By integrating Regularized PriorBand, we jointly optimize segmentation performance and training runtime, addressing practical constraints in medical settings. Our comprehensive evaluation on all ten Medical Segmentation Decathlon datasets demonstrates that Auto-nnU-Net consistently outperforms or matches strong baselines while maintaining practical resource requirements. We further analyzed the contributions of HPO and NAS through ablation studies, examined the transferability of optimized configurations across datasets, and assessed hyperparameter importance. These insights contribute to a deeper understanding of the design and optimization of our segmentation approach in diverse clinical settings. Overall, Auto-nnU-Net provides a flexible and resource-aware foundation for automated medical image segmentation, enabling robust model design under real-world constraints.

**Limitations**. This study, including results for both nnU-Net and Auto-nnU-Net, is based on the 3D U-Net architecture without post-processing or ensembling, which may not fully reflect the original nnU-Net's performance (Isensee et al., 2020a). However, incorporating ensembling — common in AutoML (Erickson et al., 2020) — would likely enhance Auto-nnU-Net's results. Regarding our Pareto analysis, lower-budget configurations approximate full-budget performance and reveal runtime-accuracy trade-offs. Lastly, while surrogate models in DeepCAVE may introduce slight approximation errors, the findings provide a strong foundation for advancing AutoML in MIS.

**Future research**. Future work could extend evaluations to the full nnU-Net pipeline and further investigate how dataset properties affect AutoML outcomes. Warm-starting AutoML with multiple default configurations (Pfisterer et al., 2018) and meta-learning (Feurer et al., 2015; Vanschoren, 2019; Aguiar et al., 2019) could improve the efficiency of AutoML for MIS. Finally, zero-shot AutoML with pre-trained models (Öztürk et al., 2022) could enhance adaptability while reducing costs.

## 8 Broader Impact Statement

AutoML for MIS can improve diagnostic accuracy and efficiency by reducing manual tuning and supporting advanced model development in collaboration with medical professionals — making it more accessible to institutions with limited ML expertise. Accurate segmentation aids early diagnosis and treatment planning, while efficient optimization is crucial in resource-constrained settings. Challenges remain, including performance dependence on training data and potential bias, which can hinder generalization. AutoML can help mitigate this by reducing expert dependence and enabling optimization across diverse datasets, promoting fairness. Future work should focus on fairness-aware methods and more efficient AutoML strategies to support ethical, sustainable deployment. If these challenges are addressed, AutoML could become a powerful tool for MIS, improving diagnostic robustness and precision while ensuring ethical and responsible deployment.

**Acknowledgements**. This work was supported by the Federal Ministry of Education and Research (BMBF), Germany, under the AI service center KISSKI (grant no. 01IS22093C). Leona Hennig and Marius Lindauer acknowledge funding by the German Federal Ministry of the Environment, Nature Conservation, Nuclear Safety and Consumer Protection (GreenAutoML4FAS project no. 67KI32007A). Steffen Oeltze-Jafra and Marius Lindauer were supported by the Lower Saxony Ministry of Science and Culture (MWK) with funds from the Volkswagen Foundation's zukunft.niedersachsen program [project name: CAIMed - Lower Saxony Center for Artificial Intelligence and Causal Methods in Medicine; grant number: ZN4257].

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

## A  Hardware and Resource Consumption

Compute nodes were equipped with the following software and hardware:

- **OS**: Rocky Linux 9.5

- **CPU**: 48xAMD EPYC 9354 32-Core Processor

- **RAM**:120 GB

- **GPU**: 1xNVIDIA H100 PCIe, 80GB VRAM, CUDA 12.4

All experiments took a total of 59 945 GPU hours, with an estimated power consumption of 0.5 kWh per GPU hour. This results in a total power consumption 29 972.5 kWh and 10 964 kg $CO_2$ equivalents based on the average energy mix of Germany in 2023[1].

## B  Related Work

### B.1  PriorBand

In MIS, architectures and hyperparameter settings designed by experts can serve as good starting points for HPO, and leveraging this knowledge may accelerate the optimization, particularly given the high computational demands of training deep learning models. However, although BO and BOHB improve the performance of HPO, they do not explicitly incorporate this knowledge.

PriorBand (Mallik et al., 2023) addresses this limitation by integrating prior configurations, e.g., expert beliefs, into the optimization process and improves the anytime performance of existing methods such as $\pi$BO (Hvarfner et al., 2022). By incorporating prior knowledge about well-performing regions in the search space, PriorBand aims to enhance the efficiency of HPO in computationally expensive scenarios with a strong prior configuration. As the work presented in this thesis largely relies on PriorBand, we provide a more detailed discussion of this approach.

An outline of PriorBand is shown in Algorithm 1. PriorBand replaces the random sampling in HB with an ensemble sampling strategy $\mathcal{E}_\pi$ (see lines 12-13), containing the following three components:

(i) **Random Sampling from $\mathcal{U}(\cdot)$.** This strategy samples hyperparameter configurations uniformly from the search space. It enables exploration of the configuration space to find promising regions without relying on the prior distribution.

(ii) **Prior-based Sampling from $\pi(\cdot)$.** This sampling strategy leverages expert knowledge about well-performing configurations. It facilitates a local search near the prior configuration using perturbation. If the preceding configuration is accurate, it accelerates the optimization process.

(iii) **Incumbent-based Sampling from $\hat{\lambda}(\cdot)$.** This strategy samples configurations around the current best-performing configuration. By exploring the configuration space locally around the incumbent configuration, it aims to refine and improve upon it. This strategy is beneficial if the prior is not accurate or useful.

Each of the three sampling components is assigned a weight that determines the probability of the respective strategy being used when sampling from $\mathcal{E}_\pi$. The weights are denoted as $p_\mathcal{U}$, $p_\pi$, and $p_{\hat{\lambda}}$. Initially, PriorBand assigns equal weights to random and prior-based sampling to ensure a balance between exploration and leveraging expert knowledge. As the optimization progresses, the probability of random sampling decreases geometrically, increasing the proportion of the other two strategies (see line 5 of Algorithm 1).

---

[1] https://tco2e.net/kwh/country/germany/

---

**Algorithm 1:** PriorBand HPO algorithm (Mallik et al., 2023)

---

**Input:** Budgets $b_{\min}$ and $b_{\max}$, reduction factor $\eta$ (default $\eta = 3$), prior $\pi$

**Output:** Incumbent configuration $\lambda^*$

1   $\mathcal{H} \leftarrow \emptyset$ ;                                    `// all observations`

2   $s_{\max} = \lfloor \log_\eta \frac{b_{\max}}{b_{\min}} \rfloor$;

3   **for** $s \in \{s_{max}, s_{max} - 1, \ldots, 0\}$ **do**

4      $r \leftarrow s_{\max} - s$;

5      $p_{\mathcal{U}} \leftarrow 1/(1 + \eta^r)$;

6      $p_\pi \leftarrow 1 - p_{\mathcal{U}}$;

7      $p_{\hat{\lambda}} \leftarrow 0$;

8      **if** *evaluated at least one config at $b_{max}$* **then**

9          $p_\pi, p_{\hat{\lambda}} \leftarrow \text{DYNAMICWEIGHTING}(\mathcal{H}, r, p_\pi)$;

10     $n \leftarrow \lceil \frac{s_{\max}}{s+1} \rceil$ configurations;

11     **for** $i \in \{1, \ldots, n\}$ **do**

12         $d(\cdot) \leftarrow$ sample strategy by $\{p_{\mathcal{U}}, p_\pi, p_{\hat{\lambda}}\}$;

13         $\lambda_i \leftarrow$ sample from $d(\cdot)$;

14     Run successive halving on the configurations $\lambda_i$ with initial budget $\eta^{-s} \cdot b_{\max}$;

15     Add observations to $\mathcal{H}$;

16     $\lambda^* \leftarrow \arg\min_{(\lambda,c) \in \mathcal{H}} c$;

17 **return** $\lambda^*$;

---

Once the first configuration is evaluated on the maximum budget, prior-based and incumbent-based sampling are weighted dynamically (see lines 8-9 of Algorithm 1). In the DYNAMICWEIGHTING subroutine, configurations are ranked by their performance, and the likelihood of the top configurations under the prior and incumbent distributions is computed. Based on these likelihoods, PriorBand dynamically adjusts the sampling weights, assigning a higher weight to the distribution that is more likely to produce well-performing configurations. The weighting ensures efficiency in the case of well-performing and robustness against bad prior configurations. Using the weights $p_{\mathcal{U}}$, $p_\pi$, and $p_{\hat{\lambda}}$, a sampling strategy is selected for each configuration individually. Based on the chosen strategy, a hyperparameter configuration is sampled. Subsequently, similar to HB, SH is called as a subroutine to efficiently allocate resources to the most promising configurations.

**Prior-based Sampling.** In PriorBand, prior-based sampling of hyperparameters is accomplished by drawing samples from a prior distribution. The type and shape of the distribution are determined by (i) the type of hyperparameter and (ii) the prior confidence provided by the user. For numerical hyperparameters, a truncated normal distribution is defined over the range of possible values, where the mean is set to the default value. The prior confidence adjusts the standard deviation, with higher confidence resulting in a lower standard deviation. Integer hyperparameters are sampled by rounding the values to the nearest integer value. For categorical hyperparameters, in contrast, the probabilities are uniformly distributed across all values except for the default, whose probability is increased in proportion to the prior confidence.

## C Approach

### C.1 Auto-nnU-Net

Our framework extends nnU-Net by (i) the `AutoExperimentPlanner`, (ii) the `CFGUNet`, and (iii) the `AutoNNUNetTrainer`. Figure 1 shows an overview of the Auto-nnU-Net framework. Unlike nnU-Net, which relies on fixed and rule-based configurations, Auto-nnU-Net introduces a hyperparameter configuration $\lambda$ and architecture $A$ as inputs alongside the dataset fingerprint. This enables a more flexible and automated experiment planning and training process, allowing models to be systematically optimized based on different hyperparameter and architecture choices. Then, Auto-nnU-Net returns the validation accuracy and training runtime as objectives. Our interface enables powerful and flexible search strategies, including multi-objective optimization, where trade-offs between performance and efficiency can be explicitly modeled.

Auto-nnU-Net extends nnU-Net with three key components that enable flexible integration of AutoML methods. The `AutoExperimentPlanner` incorporates hyperparameter configurations into the planning process to enable the optimization of architectural properties such as the number of features, normalization, activation, and dropout. To support more expressive architectural definitions through hierarchical NAS search spaces (see Section 5.4), the `CFGUNet` translates function composition representations into neural network models. Finally, the `AutoNNUNetTrainer` extends the training pipeline with dynamic hyperparameter configurations, including optimizer settings, learning rate schedules, and augmentation strategies. Together, these components provide a unified framework for optimizing both hyperparameters and architectures in MIS.

### C.2 Regularized PriorBand

An overview of the Regularized PriorBand algorithm is shown in Algoritm 2, where changes compared to the original PriorBand algorithm (see Algorithm 1) are highlighted. For a detailed outline of PriorBand, we refer to Appendix B.1. In Line 21, we apply non-dominated sorting on the set of observations, i.e., candidate configurations $P$ in the current stage of SH. The subroutine returns a list of fronts, where the first front is the actual Pareto front of $P$ and each subsequent the updated Pareto front after removing the previous front. In Lines 23-24, we iterate over all fronts and sort the configurations within the front based on their (i) crowding distance and (ii) objective cost by calling the CROWDINGDISTANCEANDACCURACYSORTING subroutine. It sorts the configurations descendingly based on their crowding distance and, in case of equal crowding distances, the cost of the primary objective, e.g., accuracy. Regularized PriorBand thereby only considers the primary objective when a Pareto front consists only of two points with equal crowding distance.

### C.3 Hierarchical NAS for U-Nets

#### C.3.1 Search Space.
In this section, we describe the construction of our hierarchical neural architecture search space using a context-free grammar (CFG) based on the work of Schrodi et al. (2023). Additionally, we extract architecture-level features as numerical and categorical pseudo-hyperparameters, reflecting architectural properties, from the function composition representation. This facilitates post-hoc analyses, offering insights into how design choices affect segmentation performance.

Since the space of allowed architectures is constrained by the shape of the input images, we dynamically generate the context-free grammar (CFG) tailored to the dataset at hand. To determine the maximum number of stages, i.e., the possible number of downsampling operations, we leverage nnU-Nets experiment planning framework. It iteratively computes the downsampled image size until the minimum feature map size of $4 \times 4 \times 4$ voxels is reached. We refer to this number as $n_{\text{stages,max}}$. Table 2 shows the search space sizes for different values of $n_{\text{stages, max}}$.

We begin with the starting symbol S. The first production rule specifies the number of stages $n_{\text{stages}}$ in the U-Net, which can take values in the range $\left[ \left\lfloor \frac{n_{\text{stages,max}}}{2} \right\rfloor, n_{\text{stages,max}} \right]$. For example, if

**Algorithm 2:** Regularized PriorBand

**Input:** Budgets $b_{\min}$ and $b_{\max}$, reduction factor $\eta$ (default $\eta = 3$), prior configuration $\pi$
**Output:** Incumbent configuration $\lambda^*$

1   $\mathcal{H} \leftarrow \emptyset$ ;                                        `// all observations`

2   $s_{\max} = \lfloor \log_\eta \frac{b_{\max}}{b_{\min}} \rfloor$;
   `// HyperBand`

3 **for** $s \in \{s_{max}, s_{max} - 1, \ldots, 0\}$ **do**

4      $r \leftarrow s_{\max} - s$;

5      $p_{\mathcal{U}} \leftarrow 1/(1 + \eta^r)$;

6      $p_\pi \leftarrow 1 - p_{\mathcal{U}}$;

7      $p_{\hat{\lambda}} \leftarrow 0$;

8      **if** *evaluated at least one config at $b_{max}$* **then**

9          $p_\pi, p_{\hat{\lambda}} \leftarrow$ DYNAMICWEIGHTING$(\mathcal{H}, r, p_\pi)$;

        `// Sampling configurations`

10      $n \leftarrow \lceil \frac{s_{\max}}{s+1} \rceil$ configurations;

11      **for** $i \in \{1, \ldots, n\}$ **do**

12          $d(\cdot) \leftarrow$ sample strategy by $\{p_{\mathcal{U}}, p_\pi, p_{\hat{\lambda}}\}$;

13          $\lambda_i \leftarrow$ sample from $d(\cdot)$;

        `// Successive halving (SH)`

14      $\mathcal{C} \leftarrow [\lambda_1, \ldots, \lambda_n]$;

15      $k \leftarrow \frac{n}{\eta}$ ;                       `// Number of configurations for next stage`

16      **for** $b \in \{\eta^{-s} \cdot b_{max}, \eta^{-(s-1)} \cdot b_{max}, \ldots, b_{max}\}$ **do**

17          $P \leftarrow \emptyset$ ;                     `// Candidates for current stage in SH`

18          **for** $\lambda \in \mathcal{C}$ **do**

19              $c \leftarrow$ EVALUATE$(\lambda, b)$ ;         `// Evaluate and return cost vector`

20              $P \leftarrow P \cup \{(\lambda, c)\}$;

21          $F_1, \ldots, F_m \leftarrow$ NONDOMINATEDSORTING$(P)$;

22          $\mathcal{C} \leftarrow [\,]$;

23          **for** $F \in \{F_1, \ldots, F_m\}$ **do**

            `// We (1) sort based on crowding distance descendingly and (2)`
               `based on 1 - DSC ascendingly`

24              $\mathcal{C} \leftarrow \mathcal{C} +$ CROWDINGDISTANCEANDCOSTSORTING$(F)$;

25          $\mathcal{C} \leftarrow [\mathcal{C}_1, \ldots, \mathcal{C}_k]$ ;               `// Take k best candidates`

26          $k \leftarrow \frac{k}{\eta}$;

27          $\mathcal{H} \leftarrow \mathcal{H} \cup P$;

28      $\hat{\lambda} \leftarrow$ GETINCUMBENT$(\mathcal{H})$;

29      $\lambda^* \leftarrow \arg\min_{(\lambda,c) \in \mathcal{H}} c_0$;

30 **return** $\lambda^*$;

$n_{\text{stages,max}} = 4$, the first production rule is defined as

$$S ::= \texttt{U-Net}(2E,\ 2D) \mid \texttt{U-Net}(3E,\ 3D) \mid \texttt{U-Net}(4E,\ 4D) \quad , \tag{1}$$

where `U-Net` is a terminal symbol. The nonterminal symbols $2E, \ldots, 4E$ and $2D, \ldots, 4D$ represent encoder and decoder modules of two, three, and four stages, respectively.

| $n_{\text{stages,max}}$ | Search Space Size |
|:---:|---:|
| 4 | 502 400 |
| 5 | 2 140 800 |
| 6 | 8 678 400 |
| 7 | 34 892 800 |

Table 2: Hierarchical NAS search space sizes based on the maximum number of stages determined by nnU-Net. Sizes are computed following the method proposed by Schrodi et al. (2023).

For the encoder the following production rules determine whether to use a convolutional or residual encoder:

$$
\begin{aligned}
2E \ ::= \ &\texttt{ConvEncoder} \left( E_{\text{Norm}} E_{\text{Nonlin}} E_{\text{Dropout}}, CEB_1, \texttt{down}, CEB_2 \right) \ | \\
&\texttt{ResEncoder} \left( E_{\text{Norm}} E_{\text{Nonlin}} E_{\text{Dropout}}, REB_1, \texttt{down}, REB_2 \right) \\
3E \ ::= \ &\texttt{ConvEncoder} \left( E_{\text{Norm}} E_{\text{Nonlin}} E_{\text{Dropout}}, CEB_1, \texttt{down}, \ldots, CEB_3 \right) \ | \\
&\texttt{ResEncoder} \left( E_{\text{Norm}} E_{\text{Nonlin}} E_{\text{Dropout}}, REB_1, \texttt{down}, \ldots, REB_3 \right) \\
4E \ ::= \ &\texttt{ConvEncoder} \left( E_{\text{Norm}} E_{\text{Nonlin}} E_{\text{Dropout}}, CEB_1, \texttt{down}, \ldots, CEB_4 \right) \ | \\
&\texttt{ResEncoder} \left( E_{\text{Norm}} E_{\text{Nonlin}} E_{\text{Dropout}}, REB_1, \texttt{down}, \ldots, REB_4 \right) \quad .
\end{aligned}
\tag{2}
$$

The terminal symbols $\texttt{ConvEncoder}$ and $\texttt{ResEncoder}$ correspond to the respective nnU-Net building blocks, while the terminal symbol $\texttt{down}$ represents the downsampling operation. Depending on the type of encoder, a sequence of convolutional encoder or residual encoder blocks is introduced. For each stage $i \in [1, n_{\text{stages}}]$, they are denoted by the nonterminals $CEB_i$ and $REB_i$ for a convolutional and residual encoder, respectively. Additionally, the nonterminal symbols $E_{\text{Norm}}$, $E_{\text{Nonlin}}$, and $E_{\text{Dropout}}$ are introduced to represent normalization, non-linearity, and dropout components.

Similar to the encoder, the decoder production rules are constructed, but with only one type of decoder:

$$
\begin{aligned}
2D \ ::= \ &\texttt{ConvDecoder} \left( D_{\text{Norm}} D_{\text{Nonlin}} D_{\text{Dropout}}, \texttt{up}, DB_1 \right) \\
3D \ ::= \ &\texttt{ConvDecoder} \left( D_{\text{Norm}} D_{\text{Nonlin}} D_{\text{Dropout}}, \texttt{up}, DB_1, \texttt{up}, DB_2 \right) \\
4D \ ::= \ &\texttt{ConvDecoder} \left( D_{\text{Norm}} D_{\text{Nonlin}} D_{\text{Dropout}}, \texttt{up}, DB_1, \ldots, DB_3 \right) \quad .
\end{aligned}
\tag{3}
$$

Here, the nonterminals $\texttt{ConvDecoder}$ with its corresponding decoder blocks $DB_i$ for stages $i \in [1, n_{\text{stages}} - 1]$ are introduced. We note that the last encoder block with index $n_{\text{stages}}$ represents the bottleneck. Thus, the decoder contains one fewer block than the encoder.

With the production rules introduced so far, we can define both the overall topology and encoder type of the U-Net. To specify the actual number of blocks per stage, the nonterminals are replaced with terminal symbols representing the block count. The possible block counts for each stage are derived from nnU-Nets default configuration. Depending on the encoder type, each stage has a fixed number of blocks, denoted as $n_{\text{CEB},i}$ and $n_{\text{REB},i}$ for the convolutional and residual encoders, respectively. Similarly, the number of blocks per stage in the decoder is denoted as $n_{\text{DB},i}$. To control the overall model size, we introduce a maximum model scale $S_{\text{max}}$. This leads to the following production rules:

$$
\begin{aligned}
CEB_i \ &::= \ \texttt{1b} \ | \ \texttt{2b} \ | \ \ldots \ \{S_{\text{max}} \cdot n_{\text{CEB},i}\}\texttt{b} \\
REB_i \ &::= \ \texttt{1b} \ | \ \texttt{2b} \ | \ \ldots \ \{S_{\text{max}} \cdot n_{\text{REB},i}\}\texttt{b} \\
DB_i \ &::= \ \texttt{1b} \ | \ \texttt{2b} \ | \ \ldots \ \{S_{\text{max}} \cdot n_{\text{DB},i}\}\texttt{b} \quad .
\end{aligned}
\tag{4}
$$

The terminal symbols $\texttt{1b}$, $\texttt{2b}$, $\ldots$ represent the number of blocks in the respective stage, with $\{S_{\text{max}} \cdot n_{\text{CEB},i}\}$ acting as a placeholder that is replaced when the CFG is constructed.

To balance search space size and expressiveness, we allow different normalization, non-linearity, and dropout configurations for the encoder and decoder. These are defined by the following production rules:

$$
\begin{aligned}
E_{\text{Norm}}, D_{\text{Norm}} &::= \; \texttt{InstanceNorm} \mid \texttt{BatchNorm} \\
E_{\text{Nonlin}}, D_{\text{Nonlin}} &::= \; \texttt{LeakyReLU} \mid \texttt{ReLU} \mid \texttt{ELU} \mid \texttt{PReLU} \mid \texttt{GELU} \\
E_{\text{Dropout}}, D_{\text{Dropout}} &::= \; \texttt{Dropout} \mid \texttt{NoDropout} \; .
\end{aligned}
\tag{5}
$$

Here, we state an examplary search space for $n_{\text{stages,max}} = 4$ and $S_{\text{max}} = 2$:

$$
\begin{aligned}
S &::= \; \texttt{U-Net}(2E,\ 2D) \mid \texttt{U-Net}(3E,\ 3D) \mid \texttt{U-Net}(4E,\ 4D) \\
2E &::= \; \texttt{ConvEncoder}\left(E_{\text{Norm}}E_{\text{Nonlin}}E_{\text{Dropout}}, CEB_1, \text{down}, CEB_2\right) \mid \\
&\qquad \texttt{ResEncoder}\left(E_{\text{Norm}}E_{\text{Nonlin}}E_{\text{Dropout}}, REB_1, \text{down}, REB_2\right) \\
3E &::= \; \texttt{ConvEncoder}\left(E_{\text{Norm}}E_{\text{Nonlin}}E_{\text{Dropout}}, CEB_1, \text{down}, \ldots, CEB_3\right) \mid \\
&\qquad \texttt{ResEncoder}\left(E_{\text{Norm}}E_{\text{Nonlin}}E_{\text{Dropout}}, REB_1, \text{down}, \ldots, REB_3\right) \\
4E &::= \; \texttt{ConvEncoder}\left(E_{\text{Norm}}E_{\text{Nonlin}}E_{\text{Dropout}}, CEB_1, \text{down}, \ldots, CEB_4\right) \mid \\
&\qquad \texttt{ResEncoder}\left(E_{\text{Norm}}E_{\text{Nonlin}}E_{\text{Dropout}}, REB_1, \text{down}, \ldots, REB_4\right) \\
2D &::= \; \texttt{ConvDecoder}\left(D_{\text{Norm}}D_{\text{Nonlin}}D_{\text{Dropout}}, \text{up}, DB_1\right) \\
3D &::= \; \texttt{ConvDecoder}\left(D_{\text{Norm}}D_{\text{Nonlin}}D_{\text{Dropout}}, \text{up}, DB_1, \text{up}, DB_2\right) \\
4D &::= \; \texttt{ConvDecoder}\left(D_{\text{Norm}}D_{\text{Nonlin}}D_{\text{Dropout}}, \text{up}, DB_1, \ldots, DB_3\right) \\
CEB_1 &::= \; \texttt{1b} \mid \texttt{2b} \mid \texttt{3b} \mid \texttt{4b} \\
CEB_2 &::= \; \texttt{1b} \mid \texttt{2b} \mid \texttt{3b} \mid \texttt{4b} \\
CEB_3 &::= \; \texttt{1b} \mid \texttt{2b} \mid \texttt{3b} \mid \texttt{4b} \\
CEB_4 &::= \; \texttt{1b} \mid \texttt{2b} \mid \texttt{3b} \mid \texttt{4b} \\
REB_1 &::= \; \texttt{1b} \mid \texttt{2b} \\
REB_2 &::= \; \texttt{1b} \mid \texttt{2b} \mid \texttt{3b} \mid \texttt{4b} \mid \texttt{5b} \mid \texttt{6b} \\
REB_3 &::= \; \texttt{1b} \mid \texttt{2b} \mid \texttt{3b} \mid \texttt{4b} \mid \texttt{5b} \mid \texttt{6b} \\
&\qquad \texttt{7b} \mid \texttt{8b} \\
REB_4 &::= \; \texttt{1b} \mid \texttt{2b} \mid \texttt{3b} \mid \texttt{4b} \mid \texttt{5b} \mid \texttt{6b} \\
&\qquad \texttt{7b} \mid \texttt{8b} \mid \texttt{9b} \mid \texttt{10b} \mid \texttt{11b} \mid \texttt{12b} \\
DB_1 &::= \; \texttt{1b} \mid \texttt{2b} \mid \texttt{3b} \mid \texttt{4b} \\
DB_2 &::= \; \texttt{1b} \mid \texttt{2b} \mid \texttt{3b} \mid \texttt{4b} \\
DB_3 &::= \; \texttt{1b} \mid \texttt{2b} \mid \texttt{3b} \mid \texttt{4b} \\
DB_4 &::= \; \texttt{1b} \mid \texttt{2b} \mid \texttt{3b} \mid \texttt{4b} \\
E_{\text{Norm}} &::= \; \texttt{InstanceNorm} \mid \texttt{BatchNorm} \\
E_{\text{Nonlin}} &::= \; \texttt{LeakyReLU} \mid \texttt{ReLU} \mid \texttt{ELU} \mid \texttt{PReLU} \mid \texttt{GELU} \\
E_{\text{Dropout}} &::= \; \texttt{Dropout} \mid \texttt{NoDropout} \\
D_{\text{Norm}} &::= \; \texttt{InstanceNorm} \mid \texttt{BatchNorm} \\
D_{\text{Nonlin}} &::= \; \texttt{LeakyReLU} \mid \texttt{ReLU} \mid \texttt{ELU} \mid \texttt{PReLU} \mid \texttt{GELU} \\
D_{\text{Dropout}} &::= \; \texttt{Dropout} \mid \texttt{NoDropout}
\end{aligned}
\tag{6}
$$

**C.3.2 Prior-based Sampling of Architectures.** In PriorBand (Mallik et al., 2023), prior-based sampling of hyperparameters is accomplished by drawing samples from a prior distribution. The type and shape of the distribution are determined by (i) the type of hyperparameter and (ii) the prior confidence provided by the user. For numerical hyperparameters, a truncated normal distribution is defined over the range of possible values, where the mean is set to the default value. The prior confidence adjusts the standard deviation, with higher confidence resulting in a lower standard deviation.

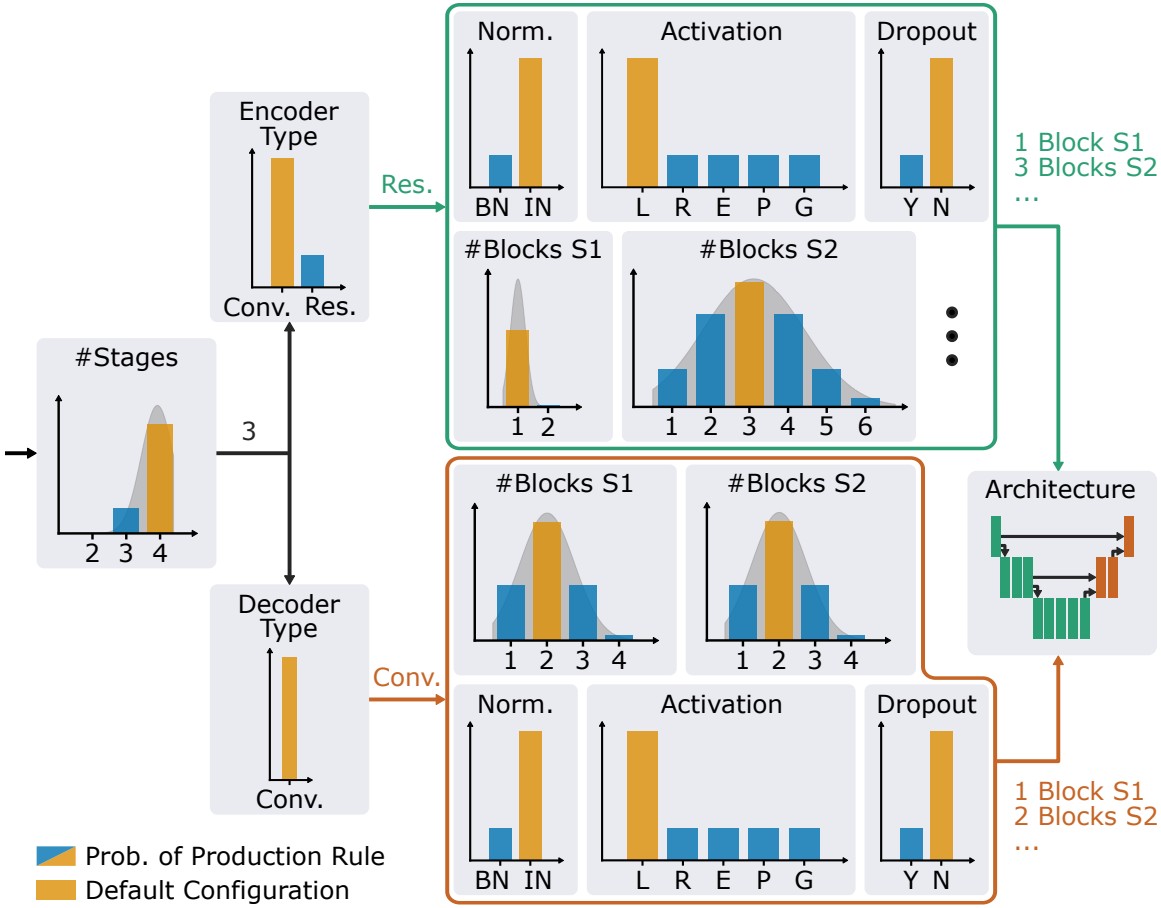

Figure 5: Overview of the prior-based sampling procedure for HNAS. Each block represents a design decision, i.e., the production rule of the CFG with its corresponding probability distribution. The probability of the possible value is indicated by the blue bar. The probability of sampling the default value is highlighted in orange. Notably, design decisions and default values differ based on previously selected values. Arrows indicate subsequent design decisions. First, the number of stages is sampled, then the encoder and decoder type are sampled. Based on the encoder type, the number of blocks per stage, as well as normalization, non-linearity, and dropout, are sampled. Abbreviations: BN = BatchNorm, IN = InstanceNorm, L = LeakyReLU, R = ReLU, E = ELU, P = PreLU, G = GeLU, Y = Yes (True), N = No (False).

Integer hyperparameters are sampled by rounding the values to the nearest integer value. For categorical hyperparameters, in contrast, the probabilities are uniformly distributed across all values except for the default, whose probability is increased in proportion to the prior confidence.

To apply this concept to hierarchical architectures, we represent design decisions as integer and categorical hyperparameters. For example, we model the type of encoder as a categorical hyperparameter with the convolutional encoder as the default value. Based on the association of production rules with probability distributions proposed by Schrodi et al. (2023), we leverage the distributions of categorical and integer hyperparameters for the production rules. Figure 5 shows an overview of the prior-based sampling within the hierarchical NAS search space. We consider an examplary search space with $n_{\text{stages,max}} = 4$ and $S_{\text{max}} = 2$. We begin by sampling the number of stages using its associated production rule, which allows the U-Net to contain two, three, or four stages. Since the default for this dataset is four, it is associated with the highest probability. Here, we consider the network to consist of three stages. Subsequently, the encoder and decoder

are sampled. In our example, we sample a residual encoder. Thus, the subsequent distributions are computed based on the default block counts in nnU-Net for a residual encoder. Here, the first stage (S1) consists of a single block, whereas the second stage (S2) comprises three blocks. For simplicity, we omit the number of blocks for the remaining stages in this example. Similarly, the remaining design decisions are sampled for the decoder.

By following this approach, we are able to dynamically produce hierarchical prior distributions based on the corresponding default configurations in different branches within the search space.

## D Experimental Setup

### D.1 Datasets

The Medical Segmentation Decathlon (MSD) (Simpson et al., 2019; Antonelli et al., 2022) is a collection of ten image segmentation datasets from the medical domain. By focusing on diversity with respect to clinical tasks, modalities, and data characteristics, the MSD aims to serve as a standard for the evaluation of image segmentation algorithms. The MSD is publicly available and provides access to all ten datasets for development and research purposes. Live ranks of submissions are stated on the challenge leaderboard[2].

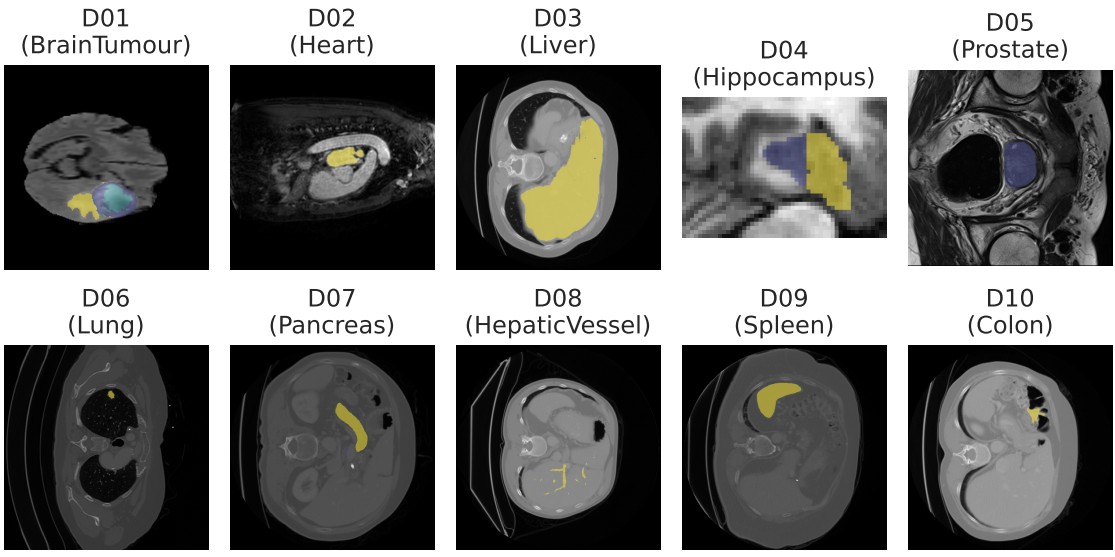

Figure 6: Example images from the MSD datasets with highlighted target labels, where each color represents an individual class. An image corresponds to the slice with the highest number of foreground voxels in the third image dimension. For 4D volumes, i.e., mp-MRI scans, the first parameter setting is selected.

**Tasks, Modalities, and Characteristics.** Figure 6 shows an overview of the ten datasets in the MSD, which we refer to as D01 to D10. Each image contains a slice of a 3D volume with the target foreground labels highlighted. For 4D volumes, the first modality is considered. The MSD tasks cover a diverse range of segmentation tasks across different anatomical regions and imaging modalities. Possible modalities are magnetic resonance imaging (MRI), computer tomography (CT), and multiparametric MRI (mp-MRI). D01[3], for instance, shows brain tumor segmentations of an

---

[2]https://decathlon-10.grand-challenge.org/evaluation/challenge/leaderboard/
[3]We use the original dataset name in British English even though this thesis is written in American English.

mp-MRI, while D03 and D06 contain segmentations of CT scans. In addition, the size and structure of segmented areas vary between datasets. D07, we see fine-grained structures for two foreground target classes, whereas D02 features larger segmented areas corresponding to a single foreground target class. This highlights the fundamental differences between datasets, which may necessitate specifically tailored models for each task to address their unique challenges and segmentation characteristics.

| Task | Name | Modality | #Images | Shape / Dimensions | #Classes |
|------|------|----------|---------|--------------------|----------|
| D01 | BrainTumour | mp-MRI | 750 | [198,169,138] / 4D | 3 |
| D02 | Heart | MRI | 30 | [115,320,232] / 3D | 1 |
| D03 | Liver | CT | 201 | [432,512,512] / 3D | 2 |
| D04 | Hippocampus | MRI | 394 | [36,50,35] / 3D | 2 |
| D05 | Prostate | mp-MRI | 48 | [20,320,320] / 4D | 2 |
| D06 | Lung | CT | 96 | [252,512,512] / 3D | 1 |
| D07 | Pancreas | CT | 420 | [93,512,512] / 3D | 2 |
| D08 | HepaticVessel | CT | 443 | [49,512,512] / 3D | 2 |
| D09 | Spleen | CT | 61 | [90,512,512] / 3D | 1 |
| D10 | Colon | CT | 190 | [95,512,512] / 3D | 1 |

Table 3: MSD datasets with their respective characteristics. Shapes are median shapes after transposing the input images based on the dataset fingerprint of nnU-Net (Isensee et al., 2020a). For mp-MRI, the fourth dimension contains the sequence of MRI scans using different parameters. The number of classes refers to the number of foreground labels.

Table 3 states the metadata of the ten tasks in the MSD, highlighting their key characteristics. An mp-MRI scan contains a sequence of MRI scans captured with different parameter settings, introducing an additional image dimension. The datasets also vary in size, resolution, and number of segmentation classes, leading to diverse challenges for evaluating MIS methods.

**Evaluation Protocol**. Each task in the MSD is divided into a fixed training and test set. Only input images are provided for the test set, and the corresponding labels are unavailable. After fitting a model on the training set, participants need to generate predictions for the test set and upload them to an online evaluation platform[4]. Since test set labels are not publicly available, the platform is the only method for evaluating a model on the test set. It then returns the test set accuracy using the Dice Similarity Coefficient (DSC) (Dice, 1945).

## D.2 AutoML Methods

**D.2.1 Additional Baseline**. Recent work on foundation models for computer vision has led to their application in the medical domain. *Segment Anything* (SAM) (Kirillov et al., 2023) is an image segmentation foundation model pre-trained on a dataset containing 1M images and 1B ground-truth segmentation masks. Unlike task-specific models, e.g., U-Nets, which need to be trained from scratch for each new segmentation dataset, the pre-training enables foundation models to generalize across diverse datasets and reduces the need for extensive labeled medical data.

MedSAM (Ma et al., 2024a), based on SAM, fine-tuned on large-scale medical imaging data, is a foundation model for MIS and can outperform task-specific models. However, as 3D image segmentations must be obtained by segmenting individual 2D slices, MedSAM achieves limited accuracy for 3D images. To overcome this limitation, MedSAM2 (Ma et al., 2024b) facilitates a transfer-learning pipeline for SAM2 (Ravi et al., 2024). SAM2 is a recent foundation model built on

---

[4]`https://decathlon-10.grand-challenge.org`

SAM for promptable image and video segmentation, trained on 35.5M masks from 50.9K videos. It replaces the vision transformer (ViT) (Dosovitskiy et al., 2021) in SAM with a hierarchical ViT (Ryali et al., 2023) and adds a memory attention module to condition the current frame on the previous one. The video segmentation capabilities of SAM2 enable MedSAM2 to represent 3D volumes as a sequence of 2D frames and produce improved 3D medical image segmentations compared to MedSAM.

SAM2 was originally designed for interactive image and video segmentation and requires a prompt, i.e., a user input, to identify the object to segment. During training, the MedSAM2 framework facilitates box prompts, which add bounding boxes around target segmentations alongside each input frame, i.e., a slice of a 3D image. During inference, the frame with the largest bounding box enclosing the target segmentation area is selected as the starting frame. The model is then prompted with two sequences of frames: one spanning from the starting frame to the first frame of the volume and the other extending from the starting frame to the last frame. The separate predictions from these sequences are aggregated to obtain the final segmentation mask. Notably, to determine the starting frame, this inference method relies on ground truth segmentations that require additional effort to obtain.

We leverage the pipeline proposed by the authors to finetune MedSAM2 on each individual MSD dataset. Due to resource limitations, we reduce the number of training epochs from 1 000 to 100, which is roughly equivalent to the training runtime of the most expensive nnU-Net configuration on D01, the dataset with the highest training runtime. Furthermore, we incorporate intermediate model evaluations on the validation split as accuracy estimates throughout the training process.

**D.2.2 PriorBand Setup**. For PriorBand, we rely on the setup proposed by Mallik et al. (2023) with the number of training epochs as HB budget. Given that the number of epochs is set to 1 000 by default in nnU-Net, we set $b_{\min} = 10$ and $b_{\max} = 1\,000$. We set the reduction factor $\eta$ to the default value of 3 as proposed by Li et al. (2017) and Mallik et al. (2023). We round budgets to full epochs. With the initial evaluation of the default configuration at the maximum fidelity, this leads to 129 evaluated configurations and a total budget of 22 000 epochs. As we continue runs within SH to reduce the computational demands, this results in a total of 18 308 trained epochs for an optimization run, excluding 5-fold cross-validation.

**D.2.3 Search Spaces**. In the following, we state details on hyperparameters. *Optimizer* can be *stochastic gradient descent with momentum* (SGD) (Goodfellow et al., 2016), Adam (Kingma et al., 2015), or AdamW (Loshchilov et al., 2019). *Momentum* is only enabled for SGD. *Learning Rate Scheduler* can use a polynomial schedule (PolyLRScheduler) (Mishra et al., 2019), cosine annealing schedule (Loshchilov et al., 2017), or no schedule at all (None). *Foreground Oversampling* defines the proportion of samples in each batch that must contain foreground segmentations. *Data Augmentation Factor* sets a multiplier that is applied to each individual data augmentation probability. When set to 0, no data augmentation is applied. *Model Scale* defines the scale by multiplying the default number of blocks per stage in the U-Net. Notably, ordinal hyperparameters are modeled as integer values mapped to actual hyperparameter values. For the encoder, the default changes based on the encoder type. *Base #Features* defines the number of features on base, i.e., the input and output stage of the U-Net. *Max. #Features* defines the maximum number of features in the bottleneck of the U-Net. When constructing the network, the number of features is doubled for each subsequent stage, but the maximum number is an upper bound. *Activation* can be *rectified linear unit* (ReLU) (Nair et al., 2010), LeakyReLU (Maas et al., 2013), *exponential linear unit* (ELU) (Clevert et al., 2016), *gaussian error linear unit* (GELU) (Hendrycks et al., 2016), and *parametric ReLU* (PReLU) (K. He et al., 2015). *Normalization* can be *batch normalization* (BatchNorm) (Ioffe et al., 2015) or *instance normalization* (InstanceNorm) (Ulyanov et al., 2016).

| Type | Hyperparameter | Type | Range / Values | Default Value |
|---|---|---|---|---|
| HPO | Optimizer | Categorical | {SGD, Adam, AdamW} | SGD |
| HPO | Momentum (SGD) | Float (log) | $[0.5, 0.999]$ | 0.99 |
| HPO | Initial Learning Rate | Float (log) | $[1 \cdot 10^{-5}, 0.1]$ | $1 \cdot 10^{-2}$ |
| HPO | Learning Rate Scheduler | Categorical | {PolyLRScheduler, CosineAnnealingLR, None} | PolyLRScheduler |
| HPO | Weight Decay | Float (log) | $[1 \cdot 10^{-6}, 1 \cdot 10^{-2}]$ | $3 \cdot 10^{-5}$ |
| HPO | Foreground Oversampling | Float | $[0, 1]$ | 0.33 |
| HPO | Loss Function | Categorical | {DiceLoss, CrossEntropyLoss, DiceAndCross-EntropyLoss, TopKLoss} | DiceAndCross-EntropyLoss |
| HPO | Data Augmentation Factor | Float | $[0, 3]$ | 1 |
| NAS | Encoder Type | Categorical | {Convolutional-Encoder, ResidualEncoderM} | Convolutional-Encoder |
| NAS | Model Scale | Ordinal | $[0.5, 1, 1.5, 2]$ | 1 |
| NAS | Base #Features | Integer | $[16, 64]$ | 32 |
| NAS | Max. #Features | Integer | $[160, 640]$ | 320 |
| NAS | Activation | Categorical | {LeakyReLU, ReLU, ELU, GELU, PReLU} NAS | LeakyReLU |
| NAS | Normalization | Categorical | {BatchNorm, InstanceNorm} | InstanceNorm |
| NAS | Dropout Rate | Float | $[0, 0.5]$ | 0 |

Table 4: HPO **(top)** and NAS **(bottom)** hyperparameters in the JAHS search space in Auto-nnU-Net.

| Hyperparameter | Type | Range / Values | Default Value |
|---|---|---|---|
| Dropout Rate | Float | $[0, 0.5]$ | 0.2 |
| Architecture | CFG-Architecture | - | - |

Table 5: Additional HNAS hyperparameters in the HPO+HNAS search space, replacing the NAS hyperparameters in the HPO+HNAS search space. The context-free grammar-based architecture (*CFG-Architecture*, Schrodi et al., 2023) defines the neural architecture using function compositions (see Section 5.4).

### D.3 Experimental Pipeline

Our *AutoNNUNet* package builds the entry point for all experiments and visualizations. For the baseline models, we rely on adaptions of the *nnunetv2* (Isensee et al., 2020a), *MedSAM* (Ma et al., 2024a), and *batchgenerators* (Isensee et al., 2020b) packages. These adaptions add support for running the frameworks on compute clusters. PriorBand and regularized PriorBand are implemented in our extension of the *Neural Pipeline Search* (NePS) (Stoll et al., 2023) framework. Our adaption of the *HyperSweeper* (Eimer, 2024) framework integrates multi-objective optimization methods. All models are trained and evaluated using 5-fold cross-validation based on the splits obtained by nnU-Net during its planning phase. Thus, we use the exact same splits for all baseline and optimization experiments.

## E  Additional Results

| Approach | nnU-Net | | | MedSAM2 | Auto-nnU-Net | Ablations | |
|---|---|---|---|---|---|---|---|
| Dataset | Conv | ResM | ResL | | | HPO | HPO + HNAS |
| D01 (BrainTumour) | 73.98 | 74.15 | 73.60 | 43.87 | **74.45** | 74.21 | 74.35 |
| D02 (Heart) | 93.39 | 93.40 | 93.26 | 87.66 | **93.53** | 93.43 | 93.39 |
| D03 (Liver) | 79.45 | **81.66** | 81.59 | 65.26 | 80.36 | 79.58 | 79.45 |
| D04 (Hippocampus) | 89.04 | 88.75 | 88.62 | 69.52 | **89.46** | 89.37 | 89.25 |
| D05 (Prostate) | 73.53 | 73.64 | 72.97 | 62.21 | 75.23 | **75.30** | 74.87 |
| D06 (Lung) | 68.33 | 68.03 | 68.58 | 68.32 | 69.19 | **71.01** | 69.73 |
| D07 (Pancreas) | 66.07 | 67.78 | 67.82 | 61.82 | 67.05 | 67.13 | **67.83** |
| D08 (HepaticVessel) | 68.31 | **68.66** | 67.67 | 45.39 | 68.31 | 68.31 | 68.31 |
| D09 (Spleen) | 96.66 | 96.76 | **97.03** | 93.87 | 96.76 | 97.02 | 96.92 |
| D10 (Colon) | 46.04 | 44.05 | 50.47 | **78.96** | 51.98 | 46.03 | 46.03 |
| **Mean** | 75.48 | 75.69 | 76.16 | 67.69 | **76.63** | 76.14 | 76.01 |

Table 6: Mean 5-fold cross-validation DSC [%] based on the nnU-Net dataset splits for baseline and AutoML incumbent configurations. The best-performing method per dataset is highlighted in **bold**.

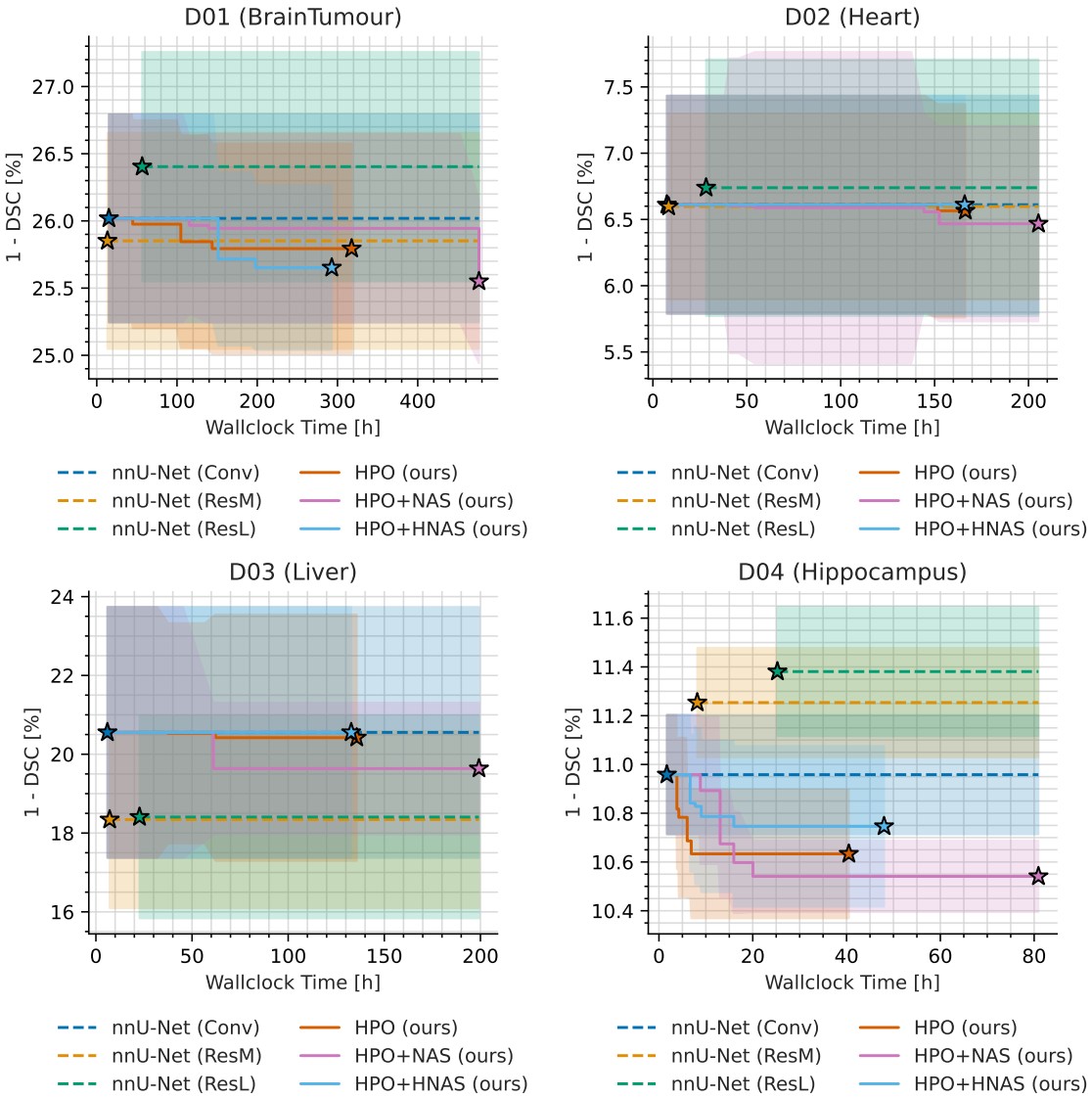

Figure 7: Optimization performance over time. Error bars indicate standard deviation across 5-fold cross-validation splits.

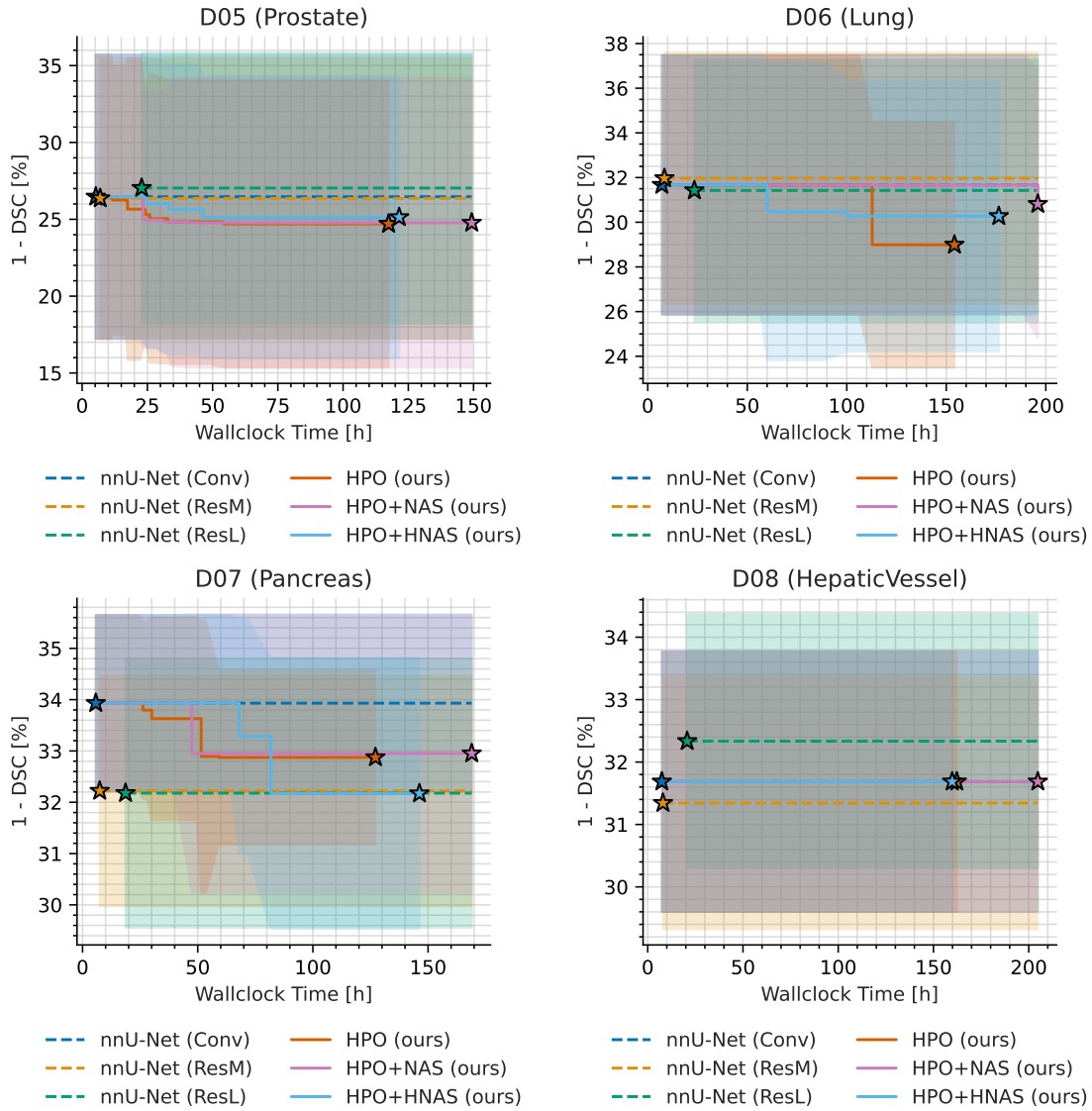

Figure 8: Optimization performance over time. Error bars indicate standard deviation across 5-fold cross-validation splits.

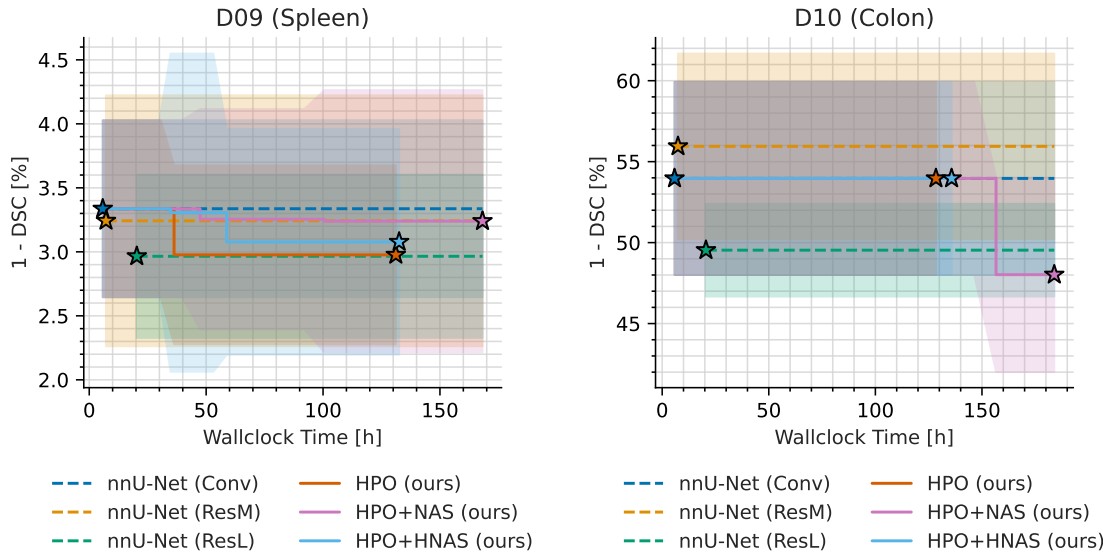

Figure 9: Optimization performance over time. Error bars indicate standard deviation across 5-fold cross-validation splits.

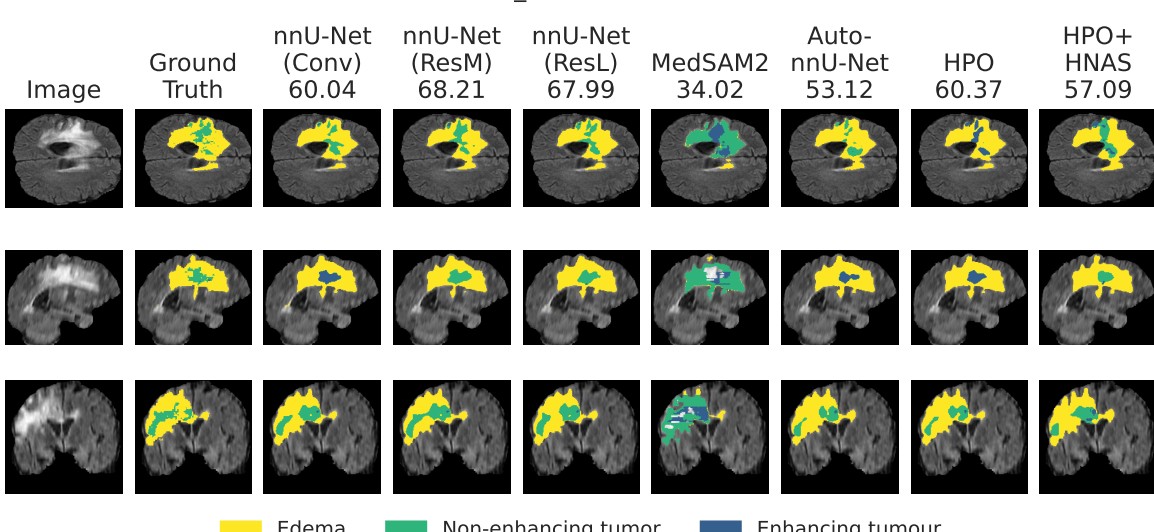

Figure 10: Qualitative segmentation results for D01. The columns correspond to the input image, ground truth segmentation mask, and predicted segmentations of the methods, where colors represent foreground classes. Numbers below the method names correspond to their respective DSC in % for this example. Each row of the figure represents a slice of the 3D volume along one axis. As this 4D volume is an mp-MRI scan, the first parameter setting is selected, yielding a 3D volume. Results for all datasets are stated in our GitHub repository.

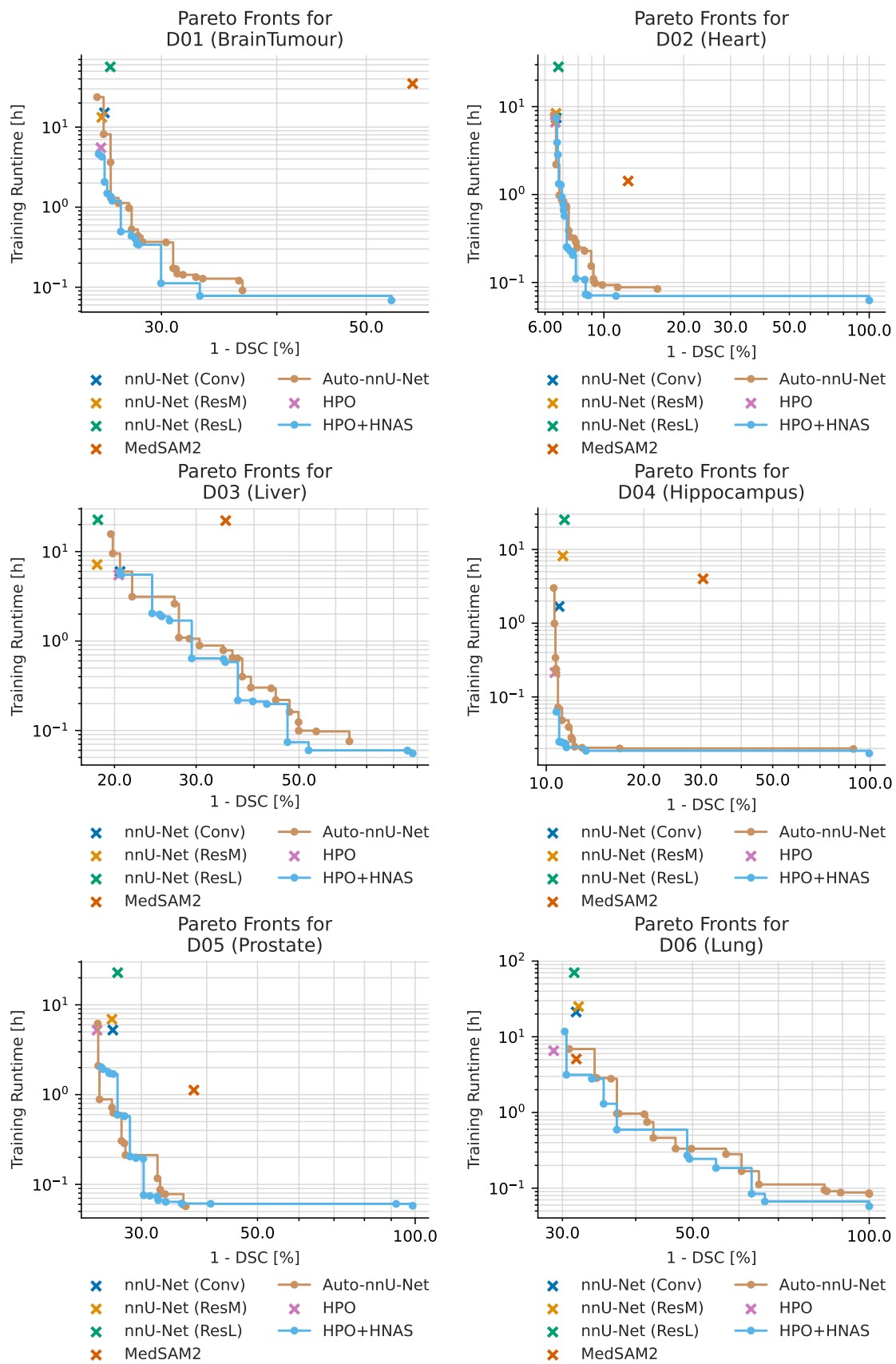

Figure 11: Pareto fronts of HPO+NAS and HPO+HNAS compared to the baselines and HPO results.

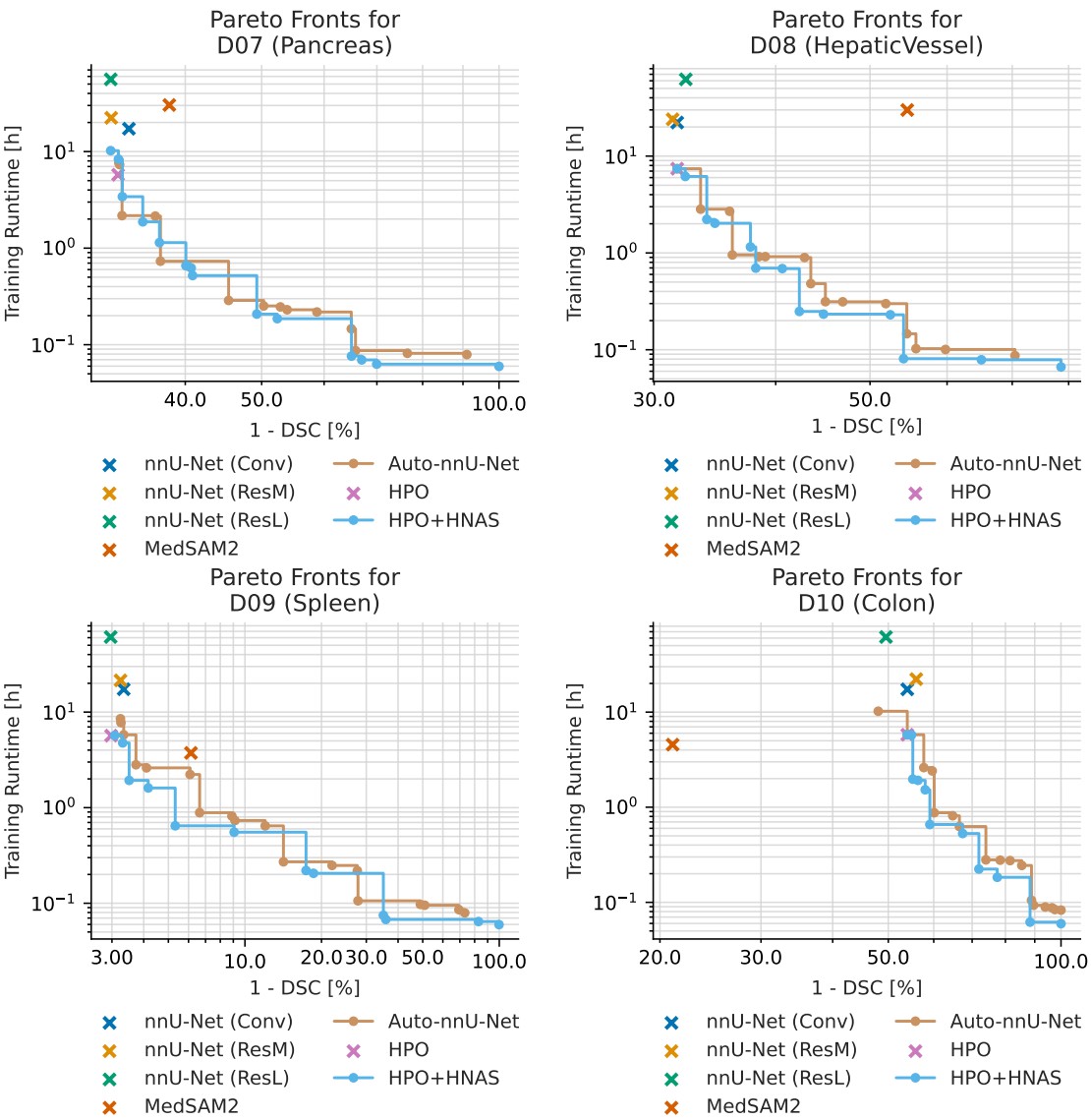

Figure 12: Pareto fronts of HPO+NAS and HPO+HNAS compared to the baselines and HPO results.

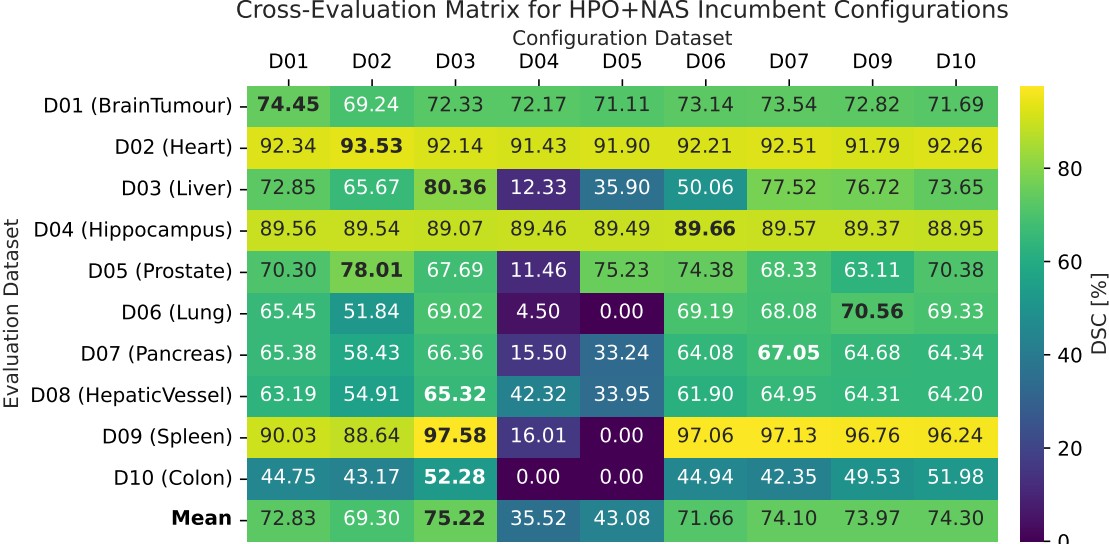

Figure 13: Cross-evaluation matrix for HPO+NAS incumbent configurations. Each cell states the 5-fold cross-validation DSC [%] when applying an incumbent configuration of a dataset (column) to a different dataset (row). In addition, the mean per incumbent configuration is stated. The highest accuracy per evaluation dataset is indicated in **bold**.

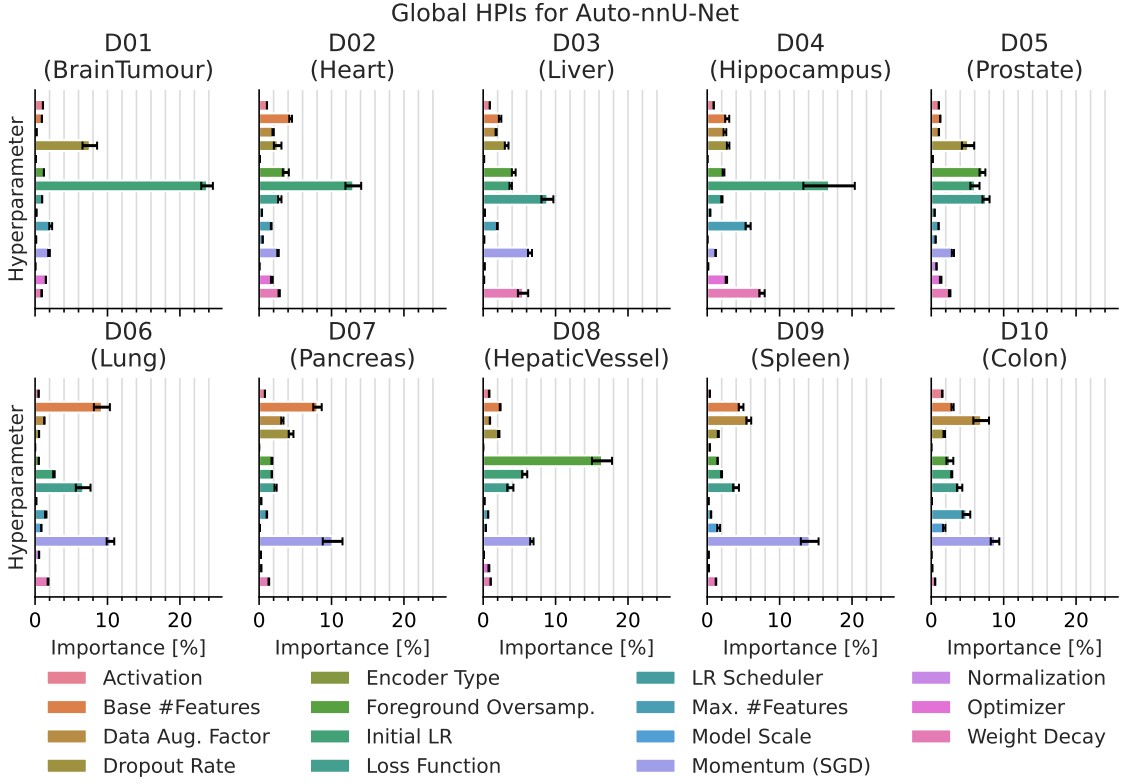

Figure 14: Global functional ANOVA (fANOVA) (Hutter et al., 2014) hyperparameter importance Auto-nnU-Net across all datasets for *1 - DSC* with error bars indicating variances.

