# OpenReview forum: "Auto-nnU-Net: Towards Automated Medical Image Segmentation"
_automl.cc/AutoML/2025/ABCD_Track — AutoML 2025 ABCD Track_

### Review · Reproducibility_Reviewer_JGvQ · 2025-04-29

**Comments To Authors:**

Overall Review: Was not able to replicate the results and run the code without errors.

Details of steps performed for the review using local config on a Linux system:
1. Tried cloning the anonymized repository - Repository not found error
2. Used the zip file submitted for review to create the directory
2. Created conda environment using python 3.10
3. Automated installation did not work, manually installed submodules using the commands provided, all installations were successful
4. Downloaded the dataset - Dataset001_BrainTumour
5. Successfully ran the convert and preprocess step for this dataset
6. Convert and preprocess for MedSam2 gave a file not found error - FileNotFoundError: [Errno 2] No such file or directory: '/Volumes/workplace/AutoNN-UNet/autonnunet/data/nnUNet_preprocessed/Dataset001_BrainTumour/splits_final.json'
7. Tried Baseline training for nnU-Net Conv, nnU-Net ResM and nnU-Net ResL on one dataset. Running the script creates some yaml files in the output directory but all of them ended up in this error -
File "/workplace/AutoNN-UNet/autonnunet/autonnunet/training/auto_nnunet_trainer.py", line 344, in get_training_transforms
    angle_x=rotation_for_DA["x"],
TypeError: tuple indices must be integers or slices, not str
8. Ran compute hyperband budgets successfully
9. Tried running Auto-nnU-Net training but ended up in this error -
  File "/home/miniconda3/envs/autonnunet/lib/python3.10/site-packages/hydra_plugins/hypersweeper/hypersweeper_sweeper.py", line 247, in run_configs
    self.launcher.params["timeout_min"] = int(optimized_timeout)
AttributeError: 'BasicLauncher' object has no attribute 'params'
10. Tried running the other training steps (HPO+HNAS Ablations) as well but found errors
11. Did not perform subsequent steps since they were dependent on a model being trained

Recommendations for Improvement -
1. Using containers such as docker will make it easier to reproduce results since all dependencies will be managed
2. Including trained models along with the submission would have made it easier to run the inference steps without the dependency of running training code
3. Since we do not have access to training clusters, testing of reproducibility artifacts with smaller data and on commonly available local systems (macOS/Linux/Windows) would have helped reproduce some of the results

**Review Confidence:**

4

**Review Rating:**

4

---

### Official Review · Reviewer_d3Z3 · 2025-05-01

**Comments To Authors:**

**Summary**:
This paper introduces an AutoML pipeline for medical image segmentation (MIS) that incorporates neural architecture search (NAS) and hyperparameter optimisation (HPO). Regularised PriorBand is also included to optimize training runtime as an additional optimisation objective. The application paper compares the proposed approach to other baselines, including MedSAM2, on the medical segmentation decathlon (a benchmark of 10 MIS datasets) and demonstrates significant improvements over other approaches.

**Strengths**:
This paper shows an actual real-world application of how AutoML can be used in an important area such as medical image segmentation. The new approach of combining NAS and HPO in addition with the runtime objective allows for general AutoML practitioners to use this approach with their respective hardware showing the general usability.
This paper is well-described and written and shows improvements over other methods, even the foundation model MedSAM2.

**Weaknesses**:
I don’t have any serious concerns of this shown application, apart from the limitations the authors already mentioned in their paper.

**Review Confidence:**

3

**Review Rating:**

8

---

### Official Review · Reviewer_gJeW · 2025-05-02

**Comments To Authors:**

Title: Auto-nnU-Net: Towards Automated Medical Image Segmentation

Summary:

This paper addresses the challenge of join hyperparameter and architecture search using nnU-Net for medical image segmentation (MIS). Typically, MIS approaches do not incorporate hyperparameter optimization into their approach, resulting in poor performance. Current work that leverages HPO for MIS does not perform NAS using nnU-Net. The authors propose the Auto-nnU-Net method, that combines multi-objective search for HPO by proposing Regularized PriorBand and NAS. Experimental evaluations are performed on 10 problems from the medical science decathlon dataset. Comparison with two baselines 3D UNet and MedSAM2 demonstrates that Auto-nnU-Net achieves similar performance (DSC score) compared to nnU-Net baselines, and outperforms on 5/10 datasets.

Overall, the paper is well-written and techniques are explained clearly. Leveraging multi-objective optimization to incorporate training time is useful, particularly for a compute-intensive task such as NAS. The results seem sound and interesting, and the Appendix is detailed though dense. The core novelty is from combining HPO with NAS, and using nnU-Net for MIS. Results with details of individual segmentation tasks are valuable.

One suggestion would be to add a couple of sentences to explicitly differentiate this work from Baldeon-Calisto et al.,2020; Lu et al., 2022) and (Yang et al., 2021).

**Review Confidence:**

3

**Review Rating:**

8

---

### Meta-Review · Area_Chair_bh2D · 2025-05-07

**Recommendation:** Accept
**Confidence:** 4

**Metareview:**

Two reviewers recommend accepting the paper. The meta-reviewer also read the paper and agrees with the assessment. The real-world application and benefit afforded by NAS in MIS are commendable. Reproducibility of the results are a concern due to some missing files and runtime errors. I strongly encourage the authors to resolve them upon acceptance of the paper.

Overall, I recommend accepting this paper.